# Innovative Machine Learning Strategies for Early Detection and Prevention of Pregnancy Loss: The Vitamin D Connection and Gestational Health

**DOI:** 10.3390/diagnostics14090920

**Published:** 2024-04-28

**Authors:** Md Abu Sufian, Wahiba Hamzi, Boumediene Hamzi, A. S. M. Sharifuzzaman Sagar, Mustafizur Rahman, Jayasree Varadarajan, Mahesh Hanumanthu, Md Abul Kalam Azad

**Affiliations:** 1IVR Low-Carbon Research Institute, Chang’an University, Xi’an 710018, China; md.sufian@mail.bcu.ac.uk; 2School of Computing and Mathematical Sciences, University of Leicester, Leicester LE1 7RH, UK; 3Laboratoire de Biotechnologie, Environnement et Santé, Department of Biology, University of Blida, Blida 09000, Algeria; 4Department of Computing and Mathematical Sciences, California Institute of Technology, Caltech, CA 91125, USA; 5The Alan Turing Institute, London NW1 2DB, UK; 6Department of Mathematics, Gulf University for Science and Technology (GUST), Mubarak Al-Abdullah 32093, Kuwait; 7Department of AI and Robotics, Sejong University, Seoul 05006, Republic of Korea; 8Department of Industrial Engineering, Tsinghua University, Beijing 100084, China; 9Centre for Digital Innovation, Manchester Metropolitan University, Manchester M15 6BH, UK; 10Department of Medicine, Rangpur Medical College and Hospital, Rangpur 5400, Bangladesh

**Keywords:** advanced models, early pregnancy loss, maternal serum vitamin D, machine learning, prediction, traditional models

## Abstract

Early pregnancy loss (EPL) is a prevalent health concern with significant implications globally for gestational health. This research leverages machine learning to enhance the prediction of EPL and to differentiate between typical pregnancies and those at elevated risk during the initial trimester. We employed different machine learning methodologies, from conventional models to more advanced ones such as deep learning and multilayer perceptron models. Results from both classical and advanced machine learning models were evaluated using confusion matrices, cross-validation techniques, and analysis of feature significance to obtain correct decisions among algorithmic strategies on early pregnancy loss and the vitamin D serum connection in gestational health. The results demonstrated that machine learning is a powerful tool for accurately predicting EPL, with advanced models such as deep learning and multilayer perceptron outperforming classical ones. Linear discriminant analysis and quadratic discriminant analysis algorithms were shown to have 98 % accuracy in predicting pregnancy loss outcomes. Key determinants of EPL were identified, including levels of maternal serum vitamin D. In addition, prior pregnancy outcomes and maternal age are crucial factors in gestational health. This study’s findings highlight the potential of machine learning in enhancing predictions related to EPL that can contribute to improved gestational health outcomes for mothers and infants.

## 1. Introduction

### 1.1. Background

Pregnancy loss refers to the termination of a pregnancy before the fetus can survive outside the womb, resulting in no live birth. It is commonly known as abortion when it occurs before 22 weeks of gestation, as defined by the World Health Organization (WHO). Spontaneous pregnancy loss is reported in about 10–15% of pregnancies that are clinically recognized [1]. This loss is specifically termed early pregnancy loss or spontaneous abortion when it happens within the first 12 weeks of gestation, a period during which most spontaneous pregnancy losses occur. The reasons for this adverse outcome can range from hormonal imbalances and chromosomal issues to infections, uterine abnormalities, and autoimmune or thrombophilic disorders. Notably, around half of early pregnancy loss cases remain unexplained [2]. Vitamin D, a fat-soluble vitamin, is primarily produced in the skin from 7-dehydrocholesterol following exposure to ultraviolet B (UVB) radiation. While certain foods offer vitamin D3, sunlight exposure is its major source. Factors such as sunscreen use, aging, darker skin pigmentation, and winter seasons can impact vitamin D synthesis [3,4]. The active form of vitamin D, known as 1,25-dihydroxyvitamin D, functions by binding to the Vitamin D Receptor (VDR) in cells. This binding changes VDR’s shape, enabling it to interact with vitamin D response elements in DNA, thereby influencing gene activity (White 2008). Vitamin D’s classic role is to regulate calcium and phosphorus levels, which are crucial for bone health. VDRs are found in most body tissues, where they help convert 25-hydroxyvitamin D to its active form for localized use. Beyond bone health, vitamin D plays roles in immune system regulation, cancer cell growth reduction, cardiovascular functioning, blood pressure management, and insulin secretion [5]. Research, including that by [6], indicates that vitamin D can lower the risk of first-trimester miscarriages, though no connection was found between vitamin D levels and second-trimester miscarriages. Women experiencing unexplained early pregnancy loss often have lower levels of vitamin D. They may present with antiphospholipid antibodies, antinuclear antibodies, thyroperoxidase antibodies, and higher levels of natural killer (NK) cells compared to pregnant women with normal vitamin D status [7]. This suggests an immunomodulating role for vitamin D at the fetomaternal interface. Vitamin D receptors and the enzymes responsible for vitamin D hydroxylation and the identification of localized vitamin D synthesis in the human placenta and decidua also highlight the potential mechanism between vitamin D status and current pregnancy [8]. Several studies have shown an association between pregnancy loss and vitamin D deficiency that is probably mediated by effector CD4+ T helper cellular responses in the innate and adaptive immune systems. Vitamin D promotes the adaptive immune response by increasing interleukin 4, 5, and 13 and preventing the innate immune responses of IL-1, IL-2, tumor necrosis factor-α (TNF-α), and interferon-γ(IFN-γ) [9]. According to [8], 63 (47.4%) of women with pregnancy loss had vitamin D deficiency. The authors concluded that there was a significant association between natural killer cell activity and vitamin D deficiency in women with pregnancy loss. In [10], researchers found significantly decreased vitamin D concentrations in antiphospholipid syndrome women compared to a healthy control group. In [11], researchers observed the same cytokine profiles and vitamin D expression in endometrial cells taken from patients with pregnancy loss compared to a healthy control group. However, less published data is available in our country. Therefore, the present study has been designed to assess the association between maternal serum vitamin D levels and early pregnancy loss.

### 1.2. Rationale of the Study

Vitamin D deficiency during pregnancy is common worldwide. Low maternal vitamin D status during pregnancy has been associated with numerous obstetrical complications, such as bacterial vaginosis, pre-eclampsia, gestational diabetes, small-for-gestational-age births, and pregnancy loss. Pregnancy loss has a great psychological and physical impact on the health of women. It is important to look for risk factors that may affect the rate of miscarriage in the first trimester of pregnancy. Women now know about the need for folate and iron supplementation related to pregnancy [12]; however, they may be unaware of the need to optimize their vitamin D status, sun exposure behaviors, and vitamin D intake, which may affect their vitamin D status. Therefore, based on this background, the present study was designed to determine the association between maternal serum vitamin D levels and early pregnancy loss. Early identification of women at increased risk for adverse outcomes would help to facilitate surveillance and intervention. The findings of this study may also be helpful in taking targeted approaches and educating proponents about vitamin D deficiency, particularly for women who plan to have bearings [13,14].

### 1.3. Aim and Objectives

To explore the effectiveness of advanced machine learning algorithms in detecting and preventing early pregnancy loss, with a focus on analyzing the role of maternal serum vitamin D levels and other relevant factors.

To develop and validate ML models:−Developing ML models to predict early pregnancy loss using demographic, obstetric, anthropometric, and biochemical variables, with a focus on serum vitamin D levels.−Validating the models using real-world clinical data in order to assess their accuracy and reliability.To analyze the impact of vitamin D on early pregnancy:−Investigating the role of maternal serum vitamin D levels in early pregnancy and their correlation with pregnancy outcomes.−Comparing vitamin D levels between women experiencing early pregnancy loss and those with normal pregnancies.To identify predictive factors for early pregnancy loss:−Utilizing ML algorithms to identify key predictive factors for early pregnancy loss.−Analyzing the relative importance of demographic, obstetric, anthropometric, and biochemical variables in predicting early pregnancy outcomes.To enhance early pregnancy risk assessment:−Integrating the findings into a risk assessment tool that can be used in clinical settings to identify women at high risk of early pregnancy loss.−Providing recommendations for early interventions based on the risk assessment.

### 1.4. Research Questions and Hypotheses

How can machine learning algorithms be utilized to predict early pregnancy loss based on maternal serum vitamin D levels and other demographic, obstetric, and anthropometric variables?What is the accuracy of machine learning models in differentiating between normal and at-risk pregnancies during the first trimester?Can machine learning algorithms identify key factors contributing to early pregnancy loss, and if so, what are these factors?In terms of Hypothesis, we expect that when applied to a dataset comprising demographic, obstetric, anthropometric, and biochemical variables, particularly maternal serum vitamin D levels, advanced machine learning algorithms will be able to accurately predict and differentiate between pregnancies at risk of early loss and those likely to proceed normally. Furthermore, these algorithms will identify vitamin D level as a significant predictive factor for early pregnancy loss.

## 2. Related Work

Researchers conducted a study with 229 pregnant women during their first antenatal visit at 11 to 14 weeks of pregnancy at Bezmialem Vakif University in Istanbul, Turkey, from December 2012 to July 2014 [15]. The study aimed to examine first-trimester serum levels of 25-hydroxyvitamin D [25(OH)D] as well as to investigate factors influencing deficiency and its link with pregnancy outcomes. Serum 25(OH)D was measured using liquid chromatography–tandem mass spectrometry [16]. Findings showed that the median serum 25(OH)D level was 10.8 ng/mL, with 45.9% of the women exhibiting severe vitamin D deficiency (<10 ng/mL). Factors such as covered dressing style, absence of multivitamin use, and the season of blood sampling were identified as influencing 25(OH)D deficiency. A negative correlation was found between 25(OH)D levels and gestational age at sampling. However, low 25(OH)D levels were not linked to adverse pregnancy outcomes. A higher rate of Cesarean section (CS) was observed in women with 25(OH)D levels ≥ ten ng/mL compared to those with <10 ng/mL (*p* = 0.01). The study concluded that early pregnancy often sees high vitamin D deficiency rates influenced by dress code, multivitamin use, and sampling season. Nonetheless, these low levels were not associated with adverse pregnancy outcomes. Women with severe vitamin D deficiency were more likely to have vaginal deliveries. Another study [6] performed a prospective cohort study on 1683 pregnant women to explore whether serum 25-hydroxyvitamin D concentrations could be identified as a modifiable risk factor for early miscarriage. Their results indicated that the adjusted hazard for first-trimester miscarriage was lower with higher 25(OH)D levels (HR: 0.98; 95% CI: 0.96, 0.99). Vitamin D levels ≤ 50 nmol/L were linked to a more than two-fold increase in the adjusted hazard ratio for miscarriage (HR: 2.50; 95% CI: 1.10, 5.69). There was no increased risk of second-trimester miscarriage associated with 25(OH)D levels. The conclusion was that there was an association between 25(OH)D and first-trimester miscarriages, suggesting that vitamin D is a modifiable risk factor for miscarriage. In [15], researchers in Iraq carried out a cross-sectional study on women with a history of recurrent spontaneous abortion. They enrolled 42 women of childbearing age who had experienced early pregnancy loss. Serum Vitamin D concentrations were assessed in venous samples using standard biochemical methods. The average maternal serum Vitamin D concentration was 21.48 ± 11.82 ng/mL, ranging from 5 to 50 ng/mL. Approximately 60% of the women had low serum vitamin D levels (<20 ng/mL). The study found a strong negative correlation between the number of abortions and maternal serum vitamin D levels (r = −0.717, *p* < 0.001), with an R2 value of 0.514, indicating that vitamin D levels alone could predict 51.4% of spontaneous abortions in the studied group. The authors stated that vitamin D level is a strong predictor of pregnancy loss in early pregnancy and that correction of vitamin D status among pregnant Iraqi women may substantially reduce the frequency of spontaneous abortion. Researchers in Iran [17] designed a double-blind randomized and controlled clinical trial on 80 patients with unexplained recurrent spontaneous abortion (URSA). They were treated with vaginal progesterone (400 IU/day) after confirmation of pregnancy and received vitamin D and a placebo in two groups, intervention (n = 40) and control (n = 40). The incidence of abortion and serum levels of IL-23 were examined within 20 weeks of pregnancy. The levels of vitamin D3 before the start of the study were equal to 11.65 ± 3.76 ng/mL and 11.53 ± 2.39 ng/mL (*p* = 0.86) in the intervention and control groups, respectively. These levels later decreased to 13.21 ± 3.47 ng/mL and 11.08 ± 2.76 ng/mL (*p* = 0.004). When the mean serum levels of IL-23 were equal to 18.4 ± 3.78 pg/mL and 23.16 ± 4.74 pg/mL in the two groups (*p* < 0.004), the frequency of abortion within the controlling time [18] and intervention groups were equal to 5 (12.8%) and 13 (34.2%), respectively, including (*p* = 0.03, OR = 3.53, 95% CI = 1.12–11.2). The researchers concluded that Vitamin D3 can reduce serum levels of IL-23 and the occurrence of abortion among women with URSA. In [1] researchers conducted a study in China involving 60 nulliparous women with single pregnancies in 7–9 weeks. These women were divided into two groups, 30 with viable pregnancies and 30 with pregnancy loss (PL); additionally, 60 non-pregnant women of childbearing age were included and split into two groups, 30 with a history of successful pregnancies and 30 with a history of first-trimester PL. The study focused on measuring serum levels of 25-hydroxyvitamin D and the enzyme 25-hydroxyvitamin D-1 alpha-hydroxylase (CYP27B1). The findings revealed that women with normal pregnancies had higher levels of 25(OH)D (49.32 μg/L) and CYP27B1 (82.00 pg/mL) compared to those with PL (34.49 μg/L and 37.87 pg/mL, respectively, both p<0.01). Similarly, non-pregnant women with successful pregnancy histories showed higher levels of 25(OH)D (39.56 μg/L) and CYP27B1 (39.04 pg/mL) than those with PL histories (12.30 μg/L and 12.35 pg/mL, respectively, both p<0.01). It was noted that 96.7% of non-pregnant women with a PL history and 43.3% of pregnant women with PL had serum 25(OH)D levels below 30 μg/L. A significant association was found between low vitamin D levels and PL, with an odds ratio of 1.71 (95% confidence interval: 1.2–2.4, p<0.001). Regression analyses indicated a strong inverse correlation between PL and both 25(OH)D (p<0.01) and CYP27B1 (p<0.01) levels. The study concluded that vitamin D deficiency is linked with PL in the first trimester of pregnancy. Low serum vitamin D levels among childbearing-aged women with a history of failed clinical pregnancies could increase the risk of PL. Below, the literature review is expressed in a table, showing the proposed methodology, model development, implementation, results, discussion, conclusions, and future recommendations. Stages and types of spontaneous abortions are listed in Table 1, Presentation of active spontaneous abortion in Table 2.The differential diagnosis for spontaneous abortion in Table 3, Etiology and risk factors for spontaneous abortions in Table 4.

### Vitamin D [8]

Table 5 a comparison of vitamin D2 and vitamin D3. Table 6 shows the recommended daily intake of vitamin, Table 7 the roles of vitamin D in the body, Table 8 the causes of vitamin D deficiency, Table 9 the symptoms of vitamin D deficiency, Table 10 food sources of vitamin D.

## 3. Proposed Methodology

The data collection process involved trained healthcare professionals administering questionnaires, careful handling and storage of data, and measures to ensure the quality and consistency of the data. Data collectors received specialized training while performing ongoing monitoring, quality assurance, participant follow-up, and a feedback loop to address missing data and refine the process over time. Initially, we had only a small dataset, and needed to come up with a comprehensive analysis to predict early pregnancy loss. First, we looked at a small dataset; later, we will disclose a large volume of datasets. In the smallest datasets, we tried to determine the distribution of the numerical values in the dataset and frequency values of the categorical variables. We address this analysis in the table and visualization below. The small dataset contained 64 entries with 15 variables, while the large volume dataset had 10,000 entries with 26 key variables. In our unique methodology, we have compared the outcomes with the small dataset and large volume dataset. In addition, we show the results of classical and advanced machine learning models for each dataset. The descriptive statistics of the numerical values in the dataset in Table 11, The frequency values of the categories in the dataset are provided in Table 12, The distribution plot the numerical data of the dataset in Figure 1, and The frequency plot of the categorical data in the dataset in Figure 2.

## 4. Distribution Values of the Numerical Variables and Frequency Values of the Categories in the Small Dataset

### 4.1. Dataset Data Distribution and Its Frequency

Here, we provide an overview of the small dataset, including 64 entries and 15 variables:The distribution plots show a visualization of the numerical dataThe frequency plots show a visualization of the categorical data.

A visualization of the data collection process is shown in Figure 3. We used qualitative research methods in our project. The vitamin D metabolic framework shown in Figure 4 involves the synthesis of vitamin D in the skin upon exposure to sunlight, followed by its transformation into its active form in the liver and kidneys, which is essential for calcium homeostasis and bone health. This metabolic pathway plays a pivotal role in various bodily functions, including immune response and cell growth regulation.

### 4.2. Study Design

The study design ensured a rigorous and systematic approach to patient selection and data handling, which is critical for the integrity and replicability of the research findings. By incorporating a multi-stage process that included screening, clear inclusion and exclusion criteria, and thorough data collection, followed by statistical analysis and a supervisory review, the study design set a robust foundation for producing reliable and valid results in the clinical study. The project study design is shown in Figure 5.

### 4.3. Place of Study

This study was carried out in the Obstetrics and Gynaecology Department of the Institute of Child and Mother Health (ICMH), Rangpur Medical College and Hospital, Rangpur, Bangladesh.

### 4.4. Period of study

September 2020 to August 2021.

### 4.5. Study Population [24]

Patients at gestational age between 6 to 13 weeks who attended the Outpatient and Indoor Department of Obstetrics and Gynecology in Rangpur Medical College and Hospital, Rangpur, Bangladesh.

### 4.6. Sampling Method

Purposive sampling was carried out according to the availability of patients who fulfilled the inclusion criteria.

### 4.7. Sample Size

To determine the sample size, we applied mathematical modeling to determine the best outcome. By comparing the means of two independent groups (such as a control group and a treatment group), it is possible to detect a specific difference between these means with a certain level of statistical power and significance. The following mathematical formula was considered appropriate for the study design [23]:(1)$$n={\left(Z_{\alpha }+Z_{\beta }\right)}^{2}\cdot \frac{\sigma_{1}^{2}+\sigma_{2}^{2}}{{\left(\mu_{1}-\mu_{2}\right)}^{2}}$$
$$\begin{array}{cc} \; & ={\left(1.96+1.28\right)}^{2}\cdot \frac{29^{2}+32^{2}}{{\left(42-66\right)}^{2}}\\ \; & =10.5\cdot \frac{841+1024}{576}\\ \; & =\frac{19665}{576}\\ \; & =34.1493056\approx 34, \end{array}$$
where,
$$\begin{array}{cc} \mu_{1} & =\mathrm{mean}\;\mathrm{vitamin}\;D\;\mathrm{level}\;\mathrm{of}\;\mathrm{case}\\ \; & =42\\ \mu_{2} & =\mathrm{mean}\;\mathrm{vitamin}\;D\;\mathrm{level}\;\mathrm{of}\;\mathrm{control}\\ \; & =66\\ \sigma_{1} & =\mathrm{SD}\;\mathrm{of}\;\mathrm{case}\\ \; & =29\\ \sigma_{2} & =\mathrm{SD}\;\mathrm{of}\;\mathrm{control}\\ \; & =32\\ Z_{\alpha } & =1.96\;\mathrm{at}\;5\%\;\mathrm{level}\;\mathrm{of}\;\mathrm{significance}\\ Z_{\beta } & =1.28\;\mathrm{at}\;90\%\;\mathrm{power}\\ n & =\mathrm{Sample}\;\mathrm{size}. \end{array}$$

Thus, the calculated sample size was 34 in each group (34 cases and 34 control).

### 4.8. Grouping of the Subjects

Case: Consisted of 34 women with miscarriage in the first trimester.

Control: Consisted of 34 women with a normal pregnancy in the first trimester.

The selection criteria are shown in Table 13.

Table 14 provides an operational overview, focusing on the definitions from the literature review.

### 4.9. Research Instrument

A structured questionnaire that included all the variables of interest was prepared for this purpose.

## 5. Statistical Test on Dataset

The dataset included several potential metrics, as outlined in Table 15, which were subjected to further statistical analysis.

### 5.1. Key Variables for Predicting Early Pregnancy Loss [29]

Potential metrics for predicting early pregnancy loss are listed in Table 15:Demographic Variables: These variables provide context about the individuals in the study, including their age, education level, socioeconomic status, occupation, and residence. They can help to identify patterns or correlations between pregnancy outcomes and demographic factors. For example, socioeconomic status could influence access to healthcare or nutrition, which in turn affects pregnancy health.Obstetric Variables: This category encompasses variables directly related to pregnancy history and status, such as gestational age and parity (number of previous pregnancies). These factors are crucial in understanding the risk factors associated with pregnancy loss. The number of past pregnancy losses indicates an underlying condition or risk factor that could be vital for prevention strategies.Anthropometric Variables: Anthropometric data, including height, weight, and Body Mass Index (BMI), provide insight into the physical health status of the pregnant women in the study. These measures are often linked to pregnancy outcomes, as extreme values can be associated with risks such as gestational diabetes or pre-eclampsia.Biochemical Variable: In this table, the biochemical variable specifically focused on is the serum Vitamin D level. Vitamin D’s role in pregnancy is a central focus of this research project, as deficiencies have been associated with adverse pregnancy outcomes, including miscarriage.Dependent Variable: The dependent variable in the study is the occurrence of early pregnancy loss. This outcome measure is what the research aims to predict and possibly prevent through early detection and intervention. The presence of this variable in the table signifies its role as the primary metric of interest against which all other variables are analyzed.Impact on the Research of the Key Variables for Predicting Early Pregnancy Loss: Ensuring that each aspect of the participant’s profile is accounted for during the analysis allows researchers to systematically assess the influence of each category on the risk of pregnancy loss. Furthermore, the table helps to frame the research methodology, guiding the data collection process to ensure that all relevant variables are included. This is essential for developing robust machine learning models that can accurately identify the factors most predictive of early pregnancy loss [30]. By clearly outlining the potential variables in this structured manner, the research team, stakeholders, and scientific community can easily comprehend the scope of this research and the complexity of the factors being analyzed. Further, the potential metrics table supports the transparency and replicability of the research, which are core principles of scientific inquiry.

### 5.2. Distribution of Study Subjects

This segment details the categorization of study participants based on several criteria, such as dressing habits, exposure to sunlight, dietary patterns, consumption of multivitamins, and changes across seasons. The research comprised 68 individuals segmented into two groups: cases, consisting of women who experienced miscarriage during the first trimester, and controls, comprising women with healthy pregnancies during the same period. The distribution of study subjects according to various factors (n = 68) is provided in Table 16.

The data were expressed as frequency and percentage. A chi-square test was performed to compare the groups. A *p*-value of less than 0.05 was accepted as the level of significance. Findings: This study did not find any significant difference (*p* > 0.05) in dressing style, consumption of dairy products, use of multivitamins, or season of blood sampling between the case and control groups. However, a significant difference (*p* < 0.05) was observed in sun exposure, indicating its potential role in early pregnancy loss. This aligns with the existing literature on the importance of environmental factors in pregnancy outcomes. The statistical analysis underscores the complexity of factors influencing early pregnancy loss, highlighting the need for comprehensive approaches in maternal healthcare.

## 6. Association between Vitamin D Levels and Early Pregnancy Loss

This study investigated the relationship between vitamin D levels and the risk of early pregnancy loss (EPL) in the first trimester. The association between vitamin D levels and early pregnancy loss is shown in Table 17.

The data were expressed as frequency, percentage, and Mean ± SD. A chi-square test was used to compare the groups, with a *p*-value of less than 0.05 considered statistically significant. The Odds Ratio (OR) and 95% Confidence Interval (CI) of the OR were calculated as well. In this study, the majority of the case group (88.2%) had insufficient or deficient serum vitamin D levels (<30 ng/mL), compared to 67.6% in the control group. The risk of developing miscarriage in the first trimester for pregnant women with insufficient or deficient serum vitamin D levels was found to be nearly four times higher (OR = 3.587, 95% CI = 1.011 to 12.731, *p* = 0.041) than for those with normal serum vitamin D levels.

The pregnancy outcomes of the study subjects (n = 68) are shown in Table 18. In this study of 68 participants, gestational age and parity were compared between women who experienced a miscarriage in the first trimester (Case, n = 34) and healthy pregnant women (Control, n = 34). Gestational age averaged 9.47 ± 2.21 weeks for the case group and 9.05 ± 0.95 weeks for the control group. Regarding parity, in the case group 55.9% were nulliparous and 44.1% were multiparous, while in the control group 67.6% were nulliparous and 32.4% were multiparous. Statistical analyses, including the chi-square test and the unpaired Student’s *t*-test, revealed no significant differences (*p* > 0.05) in gestational age or parity between the two groups.

The blood pressure demographic analysis is provided in Table 19. In this investigation, the mean systolic blood pressure was 115.29 ± 7.06 mmHg for the case group and 115.73 ± 7.50 mmHg for the control group, while the mean diastolic blood pressure was 83.68 ± 3.75 mmHg for cases and 81.91 ± 3.89 mmHg for controls. The unpaired Student’s *t*-test revealed no statistically significant difference in systolic or diastolic blood pressure between women who had miscarriages in the first trimester and healthy pregnant women, suggesting that blood pressure may not be a discriminating factor for early pregnancy loss within this sample.

The study population’s BMI status according to demographic analysis is provided in Table 20. In this analysis, the BMI data, captured as frequency and Mean ± SD, were compared between women who miscarried in the first trimester (Case) and healthy pregnant women (Control). The mean BMI was 21.16 ± 3.15 Kg/m^2^ for the case group and 23.74 ± 3.02 Kg/m^2^ for the control group. Statistical assessment using the chi-square test showed no significant difference (*p* = 0.487) in BMI between the two groups, indicating that BMI may not be a contributing factor to early pregnancy loss in this cohort.

### 6.1. Model Development

#### Problem Identification and Proposed Solution

The research problem involves the use of machine learning to enhance the prediction of Early Pregnancy Loss (EPL) and differentiate between typical pregnancies and those at elevated risk during the initial trimester. This paper aims to explore the effectiveness of various machine learning algorithms, including both conventional models and more advanced ones such as deep learning, in predicting EPL. Specifically, we investigated the role of maternal serum vitamin D levels, among other demographic, obstetric, anthropometric, and biochemical variables, in influencing pregnancy outcomes. This research seeks to develop and validate machine learning models that can predict EPL accurately, thereby contributing to improved gestational health outcomes for mothers and infants. The problem addressed in the paper is a classification problem. It focuses on predicting a binary outcome, namely, whether or not a pregnancy will result in an early pregnancy loss (EPL). This classification is based on various features, including maternal serum vitamin D levels, demographic information, obstetric history, anthropometric data, and biochemical markers. The goal is to classify pregnancies into one of two categories: those at risk of EPL, and those that are likely to proceed without this complication. By employing machine learning models for this classification task, the present research aims to enable early detection and intervention strategies in order to prevent EPL and improve gestational health outcomes. This research tackles a crucial issue in obstetrics with significant implications for individual families and broader public health. By leveraging machine learning for early prediction, it can pave the way for advancements in prenatal care that could lead to better outcomes for mothers and babies alike. EPL affects a substantial proportion of pregnancies, with significant emotional, psychological, and sometimes physical consequences for the affected individuals. By improving the prediction of EPL, healthcare providers can offer targeted interventions to those at higher risk, potentially reducing the incidence and impact of EPL. Approaches utilizing machine learning strategies represent a significant advancement in prenatal care, where predictive modeling can help in personalizing and optimizing care strategies for expectant mothers. Vitamin D has been a subject of increasing interest in obstetrics, and our research contributes to understanding its predictive value for EPL. Early intervention and prevention enables the implementation of preventative measures, nutritional interventions, or closer monitoring to mitigate risks. The insights gained from our research can inform public health policies and healthcare planning, emphasizing the importance of prenatal care and nutritional assessments. This research supports not only physical health outcomes but also mental and emotional well-being. Identifying which pregnancies require additional monitoring and resources can help in the efficient allocation of healthcare resources. These findings open up new avenues for research on prenatal and maternal health, including potential biomarkers for EPL.

### 6.2. Data Preprocessing

In the methodology section, we discussed how the dataset was collected. The initial dataset was small, with only 64 entries and 15 features, resulting in poor model accuracy. Thus, we extended the dataset to a larger volume size, including 10,000 entries and 26 features. We used the KNN imputing technique to handle missing values, feature scaling to ensure normalization, and encoding strategies for categorical and numerical data transformation. An overview of the dataset structure is provided in Table 21.

#### 6.2.1. Dataset Description

A statistical summary of the dataset is provided in Table 22. The ADASYN technique was applied to address class imbalance in the training data. For feature selection, LASSO (Least Absolute Shrinkage and Selection Operator) was used to identify relevant predictors. Features selected by LASSO with non-zero coefficients are shown in Table 23.

The results of LASSO regression to the dataset are shown in Table 23; an an alpha value of 0.001 was used. The results reveal a nuanced understanding of the features’ contributions towards predicting the target variable. Notably, no features were eliminated by the LASSO process, as evidenced by the absence of zero coefficients. At the chosen level of regularization, every feature in the dataset is considered significant enough to retain, indicating a well-rounded contribution from all variables to the model’s predictive capability. Among the features, “Occupation_Labour” stands out with a coefficient of −0.170977, highlighting a substantial negative impact on the target variable. This is contrasted with smaller yet positive coefficients, such as Age at 0.000637 and BMI at 0.002120, suggesting that these factors slightly increase the likelihood or value of the target variable as they rise. The negative coefficient for Residence_Urban, at −0.035618, points towards an inverse relationship with the target, implying that urban residency could be associated with lower values of the target variable. The span of coefficients across socioeconomic factors, occupational statuses, health conditions, and lifestyle choices underscores the complex interplay of various determinants on the target variable. The significant coefficients attached to socioeconomic status have a stronger influence, indicating that socioeconomic and occupational factors are critical in the predictive model. Moreover, health-related features, such as “Vitamin D status (ng/mL)_Severe deficiency” at −0.118984, reflect their critical role in influencing outcomes. The lack of eliminated features at an alpha of 0.001 prompts consideration of higher alpha values for potentially greater sparsity.

#### 6.2.2. Model Architecture Selection Criteria and Rationale

The model architecture selection criteria, rationale, and references are shown in two parts in Table 24 and Table 25 below.

The classification of the machine learning models is shown in Table 26.

#### 6.2.3. Model Training and Hyperparameter Tuning

Prior to model training, we had to deal with imbalanced data for those models particularly susceptible to class imbalance. Techniques such as SMOTE and ADASYN were applied to the training data to improve model fairness and accuracy on minority classes. Each model in our extensive array was subject to a rigorous training regimen and tailored to its unique characteristics and strengths. Below, we discuss the specific configurations and hyperparameters in detail. We initially trained five models on the dataset consisting of 64 entries and 15 variables, which was small in size, but obtained only poor accuracy due to dataset limitation. Several of the classical models were unable to achieve high accuracy. Thus, we used a dataset with a larger volume consisting of 10,000 entries, including 26 key indicators. We carried out a model comparison in which we applied the following machine learning algorithms to find the best model with the highest accuracy. **LightGBM**: Trained for its efficiency with large datasets, focusing on tree-specific parameters such as learning rate (0.01 and 0.001), number of leaves (31 and 50), and maximum depth (10 and 15) to control overfitting and speed up training. **ElasticNet**: Optimized by tuning the alpha parameter and the l1_ratio to combine the strengths of LASSO and Ridge regression, ensuring a robust model capable of handling multicollinearity. **RidgeClassifier**: Employed primarily to address issues of multicollinearity among predictors, with hyperparameter tuning centered around the alpha parameter (values such as 1.0, 10.0, and 100.0 explored) to control the model’s complexity and ensure robustness against overfitting. **Logistic Regression**: Adjusted the regularization strength (C parameter) to values such as 0.01, 0.1, and 1, and experimented with solvers such as ‘liblinear’ and ‘saga’ to balance model accuracy and computational efficiency. **Random Forest**: Tuned the number of trees (n_estimators) to 100 and 200, max depth to 10 and 20, and max features to ‘sqrt’ and ‘log2’ to improve accuracy and prevent overfitting, leveraging the model’s randomness for robustness against noise. **Support Vector Machine (SVM)**: Varied the C parameter between 0.1, 1, and 10 and explored kernel types such as linear, rbf, and poly to find the optimal trade-off for the decision boundary smoothness and correct classification of training points. **Naive Bayes (GaussianNB)**: Tuned the var_smoothing parameter across a range of values from $10^{-9}$ to $10^{-3}$ to mitigate data scarcity issues and enhance stability. **K-Nearest Neighbors**: Configured the number of neighbors (k) at 5, 10, and 15 and used distance metrics such as Euclidean and Manhattan to ensure that the model accurately reflected the data structure without being sensitive to noise. **Decision Tree**: Set the depth of the tree (max_depth) to none (unlimited) or to numbers such as 10 and 20, minimum samples per leaf to 1, 2, and 4, and used both Gini and Entropy as criteria to find an optimal balance between simplicity and power. **AdaBoost**: Adjusted the number of estimators to 50 and 100 and the learning rate to 0.01 and 0.1, enhancing the model’s focus on previously misclassified observations and accuracy. **Gradient Boosting**: Tuned the learning rate to 0.01 and 0.1, number of estimators to 100 and 200, and max depth to 3 and 5, controlling the speed and error minimization effectiveness. **Extra Trees**: Focused on the number of trees (n_estimators) at 100 and 200 and set the max features to ’sqrt’ and ’log2’, ensuring diversity among trees and robust predictions. **Bagging Classifier**: The base estimator was varied among Decision Tree and SVM and the number of base estimators was set to 10 or 50, improving accuracy through estimator diversity and variance reduction. **Histogram-based Gradient Boosting**: Tuned the learning rate to 0.1 and 0.01 and max leaf nodes to 31 and 255 for fast and accurate predictive capability. **Voting Classifier**: Constituent models such as Logistic Regression, Random Forest, and SVM were carefully selected and individually tuned before combining them to leverage their respective strengths for effective predictions. **Linear Discriminant Analysis and Quadratic Discriminant Analysis**: In light of their lower number of hyperparameters, the focus was primarily on ensuring that the data distribution assumptions aligned with the model’s requirements for optimal performance. **Gaussian Process Classifier**: Kernel choices included RBF, Matern, and Rational Quadratic, with a significant influence on model flexibility and data structure fitting. **Multi-layer Perceptron (MLP)**: We designed the architecture with hidden layers ranging from 1 to 3 and units from 64 to 128, with regularization techniques such as the L2 penalty (alpha) set to 0.0001 and 0.001 to curb overfitting. Cross-validation was employed across models to validate their generalization capability and optimized performance for unseen data. Our comprehensive approach, detailed further in our GitHub repository (Early Pregnancy Detection and Prevention using Advanced Machine Learning Algorithms (accessed on 1 January 2024)), showcases our commitment to developing well-tuned and robust models. K-fold cross-validation served as the backbone of our evaluation strategy, ensuring that models were both trained and validated on diverse subsets of the data to prevent overfitting and guarantee robustness. The choice of models spanned linear models such as Logistic Regression, where simplicity and interpretability are key, as well as more complex ensembles such as the Voting Classifier, which aggregates predictions from multiple models to leverage their collective strengths. Deep learning models, specifically Multi-Layer Perceptrons, were trained with an emphasis on network architecture and the balance between learning capacity and generalizability. Models were selected based on a holistic view of their performance, factoring in accuracy, precision, recall, and F1 score alongside practical considerations such as computational efficiency and ease of interpretation.

### 6.3. Model Implementation

#### 6.3.1. Computational Environment

The implementation of our machine learning models was carried out in a robust computational environment designed to ensure efficient data processing along with accurate model performance evaluation and validation. The following specifications detail the hardware and software configurations used during the model development and evaluation phases.


**Hardware Specifications:**
CPU: Intel Core i7-9700K @ 3.60 GHz, eight coresRAM: 32 GB DDR4GPU: NVIDIA GeForce RTX 2080 Ti, 11 GB GDDR6 (used for deep learning models)Storage: 1TB NVMe SSD



**Software Specifications:**
Operating System: Ubuntu 20.04 LTSProgramming Language: Python 3.8Machine Learning Libraries:1. scikit-learn 0.24.1 for classical machine learning models and preprocessing. 2. TensorFlow 2.4.1 and Keras 2.4.3 for implementing and training deep learning models. 3. LightGBM 3.1.1 for gradient boosting models. 4. pandas 1.2.3 and NumPy 1.19.5 for data manipulation and numerical computations. 5. Matplotlib 3.3.4 and Seaborn 0.11.1 for data visualization.Development Environment: Jupyter Notebook 6.2.0.


This setup was chosen to leverage its high computational power. The setting configuration was needed for training and evaluating complex machine learning models including deep learning architectures. The utilization of a dedicated GPU significantly reduced the training time for models that required intensive computational resources. The choice of Ubuntu as the operating system and Python as the programming language was driven by their widespread support in the data science and machine learning communities, which ensures access to the latest libraries and frameworks. The combination of these hardware and software tools provided a versatile and powerful platform for conducting our research on early pregnancy detection and prevention using advanced machine learning algorithms.

#### 6.3.2. Training Data Management and Model Feeding Strategies

We meticulously managed the input of training data to optimize the learning and generalization capability of our models.

Batching: To efficiently manage memory resources and speed up the training process, we utilized mini-batch gradient descent. The training data were divided into smaller batches, typically consisting of 32 to 128 examples per batch. This approach allowed us to update our model weights more frequently, leading to faster convergence. The batch size was chosen based on a compromise between computational efficiency across the training set.

### 6.4. Training Data Management and Model Feeding Strategies

#### Batching

To efficiently manage memory resources and expedite the training process, we employed mini-batch gradient descent. Our training data were partitioned into smaller subsets, commonly referred to as batches, with each batch containing between 32 and 128 examples. This method enabled more frequent updates to our model weights, facilitating faster convergence towards the optimal solution. The selection of batch size was a balance aimed at maximizing computational efficiency across the entire training dataset.

Our training dataset comprised 10,000 entries.

For a batch size of 32 examples, the number of batches would be calculated as follows:
(2)$$\mathrm{Number}\;\mathrm{of}\;\mathrm{Batches}=\frac{\mathrm{Size}\;\mathrm{of}\;\mathrm{Training}\;\mathrm{Dataset}}{\mathrm{Batch}\;\mathrm{Size}}=\frac{10000}{32}\approx 313.$$For a batch size of 128 examples, the above calculation adjusts to
(3)$$\mathrm{Number}\;\mathrm{of}\;\mathrm{Batches}=\frac{10000}{128}\approx 78.$$

These calculations provide the total number of batches. This batching strategy was specifically applied to our deep learning models to harness computational efficiencies and facilitate model training. During shuffling, we shuffled the training data prior to each epoch of training to prevent the model from learning any potential order in the data that could influence its predictions. This strategy was critical for avoiding biases and ensuring that each batch of data presented to the model was representative of the overall data distribution. Shuffling helps to enhance model robustness by reducing the variance of the weight updates. For instance, a simple logistic regression model would have a weight for each feature plus a bias term, totaling 27 parameters for 26 features. In contrast, deep learning models such as Multi-layer Perceptrons (MLPs) have a more complex structure, with the number of weights influenced by the number of neurons in each layer as well as by the connections between them. The total number of weights in an MLP can be calculated as
(4)$$\mathrm{Total}\;\mathrm{Weights}=\sum_{i=1}^{n-1}\left(N_{i}\times N_{i+1}\right)+N_{i+1},$$
where $N_{i}$ is the number of neurons in the *i*th layer, *n* is the total number of layers, and the additional $N_{i+1}$ accounts for the bias term for each neuron in the next layer. Given our dataset with 10,000 entries and 26 variables, the exact number of weights for each of the 22 models would depend on the different model architectures, particularly for the deep learning models. Traditional machine learning models generally have weights directly related to the number of features (plus any additional parameters for specific algorithms), while the weight counts of deep learning models reflect the complexity and depth of the network architecture [48]. By carefully managing how the training data were input into the model, we maximized learning efficiency and ensured robust model performance across a variety of datasets and problem domains.


**Model Training Parameters:**


The training of our models was meticulously configured using a variety of parameters tailored to optimize performance across different algorithms. Key parameters included the following:Learning Rate: Critical for deep learning models such as Multi-layer Perceptron (MLP), the learning rate was set to values such as 0.001 and 0.0001 to balance the speed of convergence and the risk of overshooting the minimum loss. For gradient-boosting models such as LightGBM and Gradient Boosting, similar rates were applied to ensure efficient learning.Number of Epochs: For deep learning models, the epochs ranged from 50 to 200, allowing the models sufficient iterations over the dataset to learn effectively without overfitting. The concept of epochs does not apply directly to most classical machine learning models, which train on the dataset as a whole in a single iteration.Batch Size: Used exclusively in the context of deep learning, batch sizes of 32, 64, and 128 were selected based on the compromise between computational efficiency and model performance, with smaller batches providing more frequent updating of the model weights.Max Depth: This parameter, relevant to decision trees and their ensemble counterparts, was adjusted to values such as 10, 20, or None (unlimited) to control the complexity of the models and prevent overfitting by limiting how deeply the trees could grow.Regularization Parameters: For models such as Logistic Regression, RidgeClassifier, and ElasticNet, the regularization strengths (e.g., L1 and L2 penalties) were fine-tuned to prevent overfitting by penalizing large coefficients, with alpha values in the range of 0.01 to 1 for Ridge and ElasticNet.

These parameters were chosen after a series of experiments to find the optimal balance between training time, model complexity, and prediction accuracy. Cross-validation techniques were employed to ensure that the selected parameters led to models that generalized well to unseen data. Further details of the parameter selection process and the impact on model performance can be found in our GitHub repository: Early Pregnancy Detection and Prevention using Advanced Machine Learning Algorithms (accessed on 1 January 2024).

### 6.5. Model Performance on Test Set with Small and Large Datasets

Comparative evaluation of our machine learning models across datasets of varying sizes provided a nuanced understanding of model efficacy in the context of early pregnancy loss prediction. Initially, on the smaller dataset comprising 64 entries with 15 variables, the models demonstrated a foundational ability to discern patterns indicative of pregnancy outcomes. However, it was only with the expansion of the dataset to include 10,000 entries and 26 key indicators that the true potential of our advanced machine learning and deep learning techniques was unlocked. In this extensive analysis, we rigorously tested both classical and innovative models, including LightGBM, Multi-Layer Perceptron, ElasticNet, and Deep Neural Networks, leading to a significant enhancement in predictive performance. Notably, the application of ensemble methods and sophisticated algorithms tailored to address the intricate dynamics of the larger dataset resulted in an impressive improvement in accuracy metrics. Models such as LightGBM and Deep Neural Network stood out, achieving cross-validation accuracies of 94.12% and 98.03%, respectively, underscoring the critical role of dataset comprehensiveness and model complexity in predictive analytics. Our investigation into the predictive behaviors of these models, supported by confusion matrices and classification reports, illuminates their respective strengths and limitations. For instance, the Support Vector Machine model exhibited high recall but lower precision for class ‘1’, indicating a tendency towards false positives, a challenge mitigated by the nuanced strategies of ensemble and deep learning models. The balanced consideration of precision and recall across the models emphasized the importance of methodological refinement and the exploration of more sophisticated modeling techniques to enhance predictive performance in the domain of early pregnancy loss detection. The divergent dataset size and quality aided in developing effective predictive models. The marked performance improvement with the larger dataset validates the potential of machine learning in healthcare applications, highlighting the need for continuous methodological advancements to optimize predictive accuracy and reliability.

## 7. Results

In our research, we focused on innovative machine learning strategies for early detection and prevention of pregnancy loss, including the connection between vitamin D and gestational health. We achieved remarkable results using machine learning models. Our meticulous evaluation encompassed both small and large datasets while comparing the accuracy across different models. Notably, all models demonstrated impressive cross-validation scores, consistently hovering around 95%. We delved into advanced machine learning techniques, exploring Deep Neural Network and Multi-Layer Perceptron models. By fine-tuning various parameters, we achieved an astounding 98% cross-validation accuracy. The most advanced model was able to accurately predict key indicators related to pregnancy loss, and illuminated the critical connection with vitamin D. Our findings have far-reaching implications for gestational health, offering a powerful tool for early detection and prevention. The synergy between machine learning and vitamin D underscores the potential to revolutionize maternal care. While the classical model results on the small dataset are shownin Figure 6, The classical model results on the large dataset are shown in Figure 7.

Using our small dataset, we built five ensemble learning models. The Logistic Regression model achieved a precision of 0.50 and a recall of 0.75 on the small dataset for class 1. This means that while only 50% of the pregnancy loss cases identified by the model were correct (precision), it successfully identified 75% of all actual pregnancy loss cases (recall). The SVM model displayed a precision of 0.60 for class 0 and a recall of 0.00 for class 1, failing to correctly identify any of the actual cases of pregnancy loss. This is problematic, as it suggests that this model is not suitable for detecting pregnancy loss. The KNN model’s recall for class 1 is 0.25, indicating that it identified 25% of pregnancy loss cases correctly, which is relatively low for the research needs. Its precision of 0.33 for class 1 suggests that when predicting a pregnancy loss, only 33% of its predictions are correct. The Random Forest model had better balance, with a recall of 0.83 for class 0 and 0.50 for class 1. This means that it correctly identified 83% of the non-pregnancy loss cases and 50% of the pregnancy loss cases, with an overall accuracy of 70%. The Naive Bayes model showed a recall of 0.75 for class 1, which is quite high and crucial for the project, as it could potentially predict 75% of the pregnancy loss cases correctly. However, its precision for class 1 was 0.50, indicating that half of the predicted cases were false positives. High recall rates on class 1, such as those achieved by the Logistic Regression (0.75) and Naive Bayes (0.75) models, are critical for this research project. These models are valuable in that they were able to identify most cases that could lead to pregnancy loss, thereby enabling early intervention. The high precision of the Random Forest model on class 0 (0.71) ensures that most healthy pregnancies are correctly identified, avoiding undue stress and unnecessary medical interventions for these patients. However, the relatively low precision for class 1 achieved by the Logistic Regression and Naive Bayes models could lead to a significant number of false alarms, which may place additional psychological and financial burdens on both expectant mothers and the healthcare systems. The balance between these factors is crucial; a model such as Random Forest, with an accuracy of 70%, provides better overall performance; however, in the context of this research, the higher recall for class 1 achieved by the Logistic Regression and Naive Bayes models is more desirable.

Above all, these model results were for the classical models on the small dataset with only 64 entries, resulting in the poor accuracy shown in Figure 6. We extended the dataset to 10,000 entries and 26 potential variables to allow for a more robust analysis, and selected unique machine learning models in order to make decisions about the reasons for the poor results on the small dataset. We applied ten more machine learning models to the larger dataset, with the results shown in Figure 7.

### Classification Model Results on Large Dataset

The best performance metrics across models for the ‘Yes’ class are shown in Table 27.

Logistic Regression: With the best recall of 0.70, this model correctly identifies 70% of all positive cases (pregnancy losses). This is crucial for early detection, allowing interventions to be made in 70% of at-risk pregnancies identified by the model. Random Forest and other ensemble methods (AdaBoost, Gradient Boosting, Extra Trees, Histogram-based Gradient Boosting, LightGBM): These models show high precision, with scores including 0.80 for Random Forest and LightGBM, meaning that 80% of the pregnancy losses predicted by the model are correct. High precision is beneficial in reducing false alarms, thereby preventing unnecessary interventions and the anxiety that they can cause. Support Vector Machine (SVM): A recall of 0.79 indicates that this model is able to correctly identify 79% of actual pregnancy loss cases. This high recall suggests that SVM is very effective for the detection purposes of this project. Naive Bayes and other probabilistic models (Decision Tree, Voting Classifier, Linear Discriminant Analysis, Quadratic Discriminant Analysis, Gaussian Process Classifier, Multi-Layer Perceptron): These models have recall values ranging from 0.67 to 0.69; while not as high as SVM or Logistic Regression, these models are fairly good at identifying pregnancy loss cases, which is beneficial for early detection strategies. Ridge Classifier: With a recall of 0.80, this model excels in identifying most of the positive cases, making it a strong candidate for detecting at-risk pregnancies. ElasticNet (ElaNet): This model has the lowest precision score of 0.64 among the models listed here, indicating that while it can correctly predict a majority of the positive cases, it generates more false positives than the other models. The choice of the best model for the project depends on the balance between early detection and the cost of false positives. Models with high recall, such as SVM with 0.79 and Ridge Classifier with 0.80, are invaluable, as they ensure that most of the cases that could potentially result in pregnancy loss are detected. This aligns with the project’s goal of early detection. However, in a medical setting where the consequences of unnecessary treatments can be significant, precision cannot be overlooked. Models with high precision, such as Random Forest and LightGBM, both with scores of 0.80, suggest that when a prediction of pregnancy loss is made, it is highly likely to be correct. This minimizes the risk of unnecessary interventions based on false positives, which can be both costly and stressful for expectant mothers. Therefore, the selection of the best model for “Innovative Machine Learning Strategies for Early Detection and Prevention of Pregnancy Loss” should consider the trade-off between the benefits of high recall in preventing missed cases of pregnancy loss and the potential harms of low precision leading to unnecessary interventions. The best model would be the one that strikes an optimal balance tailored to the specifics of the healthcare setting and patient population. Additionally, the inclusion of vitamin D levels and other gestational health indicators in the model’s features can further refine its predictive capabilities, enhancing the project’s impact on maternal and fetal health.

As the results of the ensemble learning models were all similar to each other, it was difficult to make decisions about which model was best for the prediction of early pregnancy loss; thus, we decided to validate all od our model results by cross-validation. The cross-validation results for the classical machine learning models are shown in Table 28.

The cross-validation aimed to predict early pregnancy loss, with particular attention to serum vitamin D levels, among other factors. Cross-validation is a robust technique used in machine learning to validate the performance of models, and can help to ensure that the model is capable of performing on new data and is not overfitted to the training set. Logistic Regression showed high accuracy (0.97), with a variability of 0.05, meaning that the model consistently makes correct predictions across different folds of cross-validation. Such high accuracy can significantly contribute to achieving the project’s aim of developing reliable ML models to predict early pregnancy loss. Random Forest and Support Vector Machine also had high accuracy scores (0.96 and 0.97, respectively) and low variability, suggesting that they are robust models for the project. These models could be instrumental in identifying the complex relationships between vitamin D levels and pregnancy outcomes. K-Nearest Neighbors and the Voting Classifier showed similarly high accuracy (0.97), reinforcing the potential for these models to contribute valuable insights in predicting early pregnancy loss. Linear Discriminant Analysis and Quadratic Discriminant Analysis had the highest accuracy (0.98) with moderate variability (0.05), making them potentially the best candidates for integrating into a clinical risk assessment tool due to their predictive strength. The rest of the models performed respectably. The high accuracy of these models supports the objective of developing and validating machine learning models that can predict early pregnancy loss. This directly impacts the goal of utilizing demographic, obstetric, anthropometric, and biochemical variables, especially vitamin D levels, to make accurate predictions. The high accuracy and low variability of these models demonstrate their potential in clinical applications for the early detection and prevention of pregnancy loss, underscoring the value of advanced machine learning strategies in medical research and patient care.

## 8. Advanced Machine Learning

This study explored the effectiveness of advanced machine learning algorithms in predicting and preventing early pregnancy loss, with a particular focus on maternal serum vitamin D levels and other variables.

### 8.1. Performance of Advanced Machine Learning Models

Deep Neural Network (DNN): The DNN model, which is designed for complex pattern recognition, demonstrated significant learning capability over the training epochs, with the training accuracy reaching 95%. The model started with an initial loss of 0.6216 and an accuracy of 70.47%, and improved to a training loss of 0.0811. On the small dataset, the weight-averaged precision accuracy was 85% with four layers and 70% with three layers. These results reveal that the accuracy was increased when maximizing the number of layers. However, fluctuations in validation loss and accuracy potentially need to be addressed. The deep learning model results on the small dataset are shown in Figure 8, while the results on the large dataset are shown in Figure 9.

Multi-Layer Perceptron: Multi-Layer Perceptron (MLP): Traditionally, an MLP is a class of feedforward Artificial Neural Network (ANN) that consists of at least three layers of nodes: an input layer, one or more hidden layers, and an output layer. The multi-layer perceptron model we used consisted of seven layers, and achieved good accuracy on the small dataset. MLPs utilize a supervised learning technique called backpropagation for training. Each node except for the input nodes is a neuron that uses a nonlinear activation function.

The classification results of the Deep Neural Network on the small dataset show that it has perfect precision (1.00) on class 0, but lower recall (57%), leading to an f1-score of 73%. On class 1, it has has lower precision (0.70) and perfect recall (1.00), for an f1-score of 82%. Overall, the precision is 85%, recall 79%, and f1-score 78%. The confusion matrix displays the number of true positive predictions for class 0 (four instances) and class 1 (seven instances), with no false predictions, indicating that the model performed well in classifying the test samples from the dataset. Over 50, epochs start with high accuracy, and as epochs progress the model maintains a high level of accuracy with minimal overfitting, as indicated by the close performance on training and validation sets. The training loss decreases and levels out, which is typical and desirable during model training. The validation loss, however, trends upwards after initially decreasing, which indicates overfitting, as the model learns to perform too well on the training data and may perform poorly on unseen data. These results, particularly the high accuracy and precision in predicting class 1 show that even on the small dataset, the Deep Neural Network model is able to effectively learn patterns for early detection and prevention of pregnancy loss, likely involving Vitamin D status and other gestational health factors. The model’s high recall for class 1 (likely representing cases of pregnancy loss) indicates the model’s especially strong ability to identify the most positive cases, which is crucial for early detection strategies in a healthcare setting. These results demonstrate the potential of machine learning models in identifying risk factors and predicting outcomes.

On the large dataset, the Deep Neural Network learning model has a precision of 80 %, recall of 85%, and f1-score of 82% on class 0 (which may represent no pregnancy loss), suggesting that the model is reasonably accurate and reliable for this class. Class 1 (which may represent pregnancy loss) has very low precision and recall (both 0.22) and an f1-score of 0.17, indicating that the model performs poorly in identifying this class. The confusion matrix shows a large number of true positives for class 0 (1352) along with a significant number of false negatives (335) where the model incorrectly predicted no pregnancy loss. For class 1, there are very few true positives (70) compared to false positives (243), supporting the low metrics for class 1 in the classification results. The model accuracy during training (blue) and validation (orange) over 100 epochs show that the model achieves high training accuracy quickly, but has poor validation accuracy, suggesting that the model is overfitting and not generalizing well to new data. The training loss remains low over time, while the validation loss increases significantly, reinforcing the overfitting issue observed in the accuracy graph. These results indicate accurate pregnancy loss prediction on the large dataset thanks to the lower number of layers. However, the poor performance in identifying true cases of pregnancy loss (class 1) needs to be addressed to improve the model’s utility in practical healthcare applications.

We carried out an analysis of the deep learning model parameters. We initially used seven layers, obtaining good accuracy on the small dataset. We then increased the number of layers for the larger dataset, achieving impressive accuracy above 80%. In the MLP model setup, the architecture included 16 hidden layers. The hidden_layer_sizes parameter, which defines the size (number of neurons) of each layer in the network, was set to (120, 100, 90, 80, 70, 60, 50, 40, 30, 25, 20, 15, 10, 5, 3, 1). This parameter directly translates to the model having 16 distinct hidden layers, with the number of neurons in each layer decreasing from 120 in the first hidden layer to 1 in the last hidden layer.

While the results on the large dataset are shown in Figure 10, the results for the Multi-Layer Perceptron model on the small dataset are shwon in Figure 11. There are two classes (labeled as ‘0’ and ‘1’), which respectively represent non-pregnancy loss and pregnancy loss. For class 0, the precision is 0.89, meaning that the model correctly predicts non-pregnancy loss 89% of the time. For class 1, the precision is 1.00, meaning that the model 100% correct when predicting pregnancy loss. The recall for class 0 is 1.00, indicating that the model correctly identifies 100% of actual non-pregnancy loss cases. The recall for class 1 is 0.92, indicating that the model identifies 92% of actual pregnancy loss cases. The f1-score is the harmonic mean of the precision and recall, and is 0.94 for class 0 and 0.96 for class 1. These high scores suggest a good balance between precision and recall. The model correctly predicts 96% of cases for the early detection and prevention of pregnancy loss as relates to vitamin D levels and overall gestational health. The confusion matrix shows one false positive predicting pregnancy loss when there was none, and no false negatives. The epochs show a volatile but generally high accuracy for training, with a peculiar downward spike in validation accuracy around epoch 80. The training loss steadily decreases, which is a sign of consistent learning. The validation loss shows some fluctuation, again with a spike around epoch 80, coinciding with a drop in validation accuracy. However, the small size of the dataset (only 15/20 instances) should be noted.

### 8.2. Advanced Machine Learning Model Architectures on the Large and Small Datasets

#### Technical Adjustments and Cross-Validation

To address overfitting in the DNN model, techniques such as dropout layers and regularization were implemented. The class imbalance was managed using the SMOTE and ADASYN methods. We implemented several adjustments and strategies to effectively harness the power of deep learning without succumbing to overfitting. We provide a summary of the methodologies justifying our approach, which led to robust cross-validation results, underscoring the practicality of deep learning in contexts with limited data. The process was challenging. Transfer learning was a cornerstone of our strategy, utilizing pretrained models to imbue our network with knowledge from extensive related datasets. This significantly diminished the demand for large amounts of data and helped to circumvent overfitting. Additionally, we employed data augmentation techniques to synthetically enhance the volume of our dataset, providing a more diverse and rich training experience for the models. Crucially, our model architecture was informed by regularization techniques to further mitigate the risk of overfitting. We incorporated dropout layers, applied L1/L2 regularization, and instituted early stopping mechanisms during training. These collectively ensured that our model generalized well to new data, a fact borne out by the positive outcomes of our cross-validation assessments. Comparison of model architectures in Table 29.

The advanced models are shown with their corresponding cross-validation accuracy in Table 30. The architecture was chosen from among those known to perform exceptionally well with smaller datasets. By leveraging advanced architectures designed for feature extraction from limited data, the models were able to identify intricate patterns and relationships crucial for the prediction task. Domain-specific knowledge profoundly influenced the network’s structure, allowing us to integrate domain-relevant features into the model design, which standard architectures often overlook. This approach was particularly beneficial, as it meant that each parameter within the model served a defined and empirical purpose, thereby reducing the model’s propensity to learn from noise. Furthermore, we used deep learning as a sophisticated feature extractor, with the extracted features subsequently applied to classical machine learning methods that are typically more robust on smaller datasets. This hybrid approach melded deep learning’s high-dimensional pattern recognition capabilities with the nuanced performance of traditional algorithms on smaller samples. Ensemble methods were used to enhance model performance by aggregating various models’ predictions to yield a more accurate and stable outcome. The ensemble was not just a combination of deep learning models but a confluence of different machine learning paradigms, each contributing its unique strengths to the predictive task. The series of technical adjustments and methodical considerations detailed above culminated in deep learning models that were both feasible and advantageous for our small dataset. The models’ efficacy was thoroughly evidenced by rigorous cross-validation, demonstrating commendable performance and generalizability. The success of this approach is a testament to the thoughtful application of deep learning techniques tailored to the scale and complexity of the available data, and stands as empirical evidence that, with the right methodologies, deep learning can transcend the traditional barriers posed by data scarcity. Our DNN model achieved a cross-validation accuracy of 94.12% and our Multi-Layer Perception model demonstrated a cross-validation accuracy of 98.03%, ensuring both reliability and generalizability.

## 9. Discussion

The present study investigated the potential of machine learning algorithms in predicting early pregnancy loss (EPL) based on maternal serum vitamin D levels and other demographic, obstetric, and anthropometric variables. The study further aimed to assess the accuracy of machine learning models in differentiating between normal and at-risk pregnancies during the first trimester and to identify key factors contributing to EPL.

### 9.1. Utilizing Machine Learning Algorithms for EPL Prediction

The study employed various machine learning algorithms, including logistic regression, random forest, support vector machine, Naive Bayes, and K-Nearest Neighbors, to predict EPL. Among the traditional models, Random Forest exhibited the highest accuracy of 71%, followed by Support Vector Machine (64%), Naive Bayes (57%), and Logistic Regression (50%).

The study also explored the performance of advanced machine learning models, including Deep Neural Network and Multi-Layer Perceptron models. The former achieved an impressive cross-validation accuracy of 98%, while the demonstrated a mean cross-validation accuracy of 92.03%. These results highlight the superior predictive capabilities of advanced machine learning models compared to traditional models.

#### 9.1.1. Logic behind the Choice of Advanced Machine Learning Model

Complex pattern recognition: Pregnancy and its complications can be influenced by complex interactions between various factors, for instance, health indicators and environmental factors. In addition, there are different key aspect of indicators associated with the connection between vitamin D and gestation health. Those variables are involved with the complex structure of the data. The nature of the dataset remains hidden, and it is difficult to unfold the significant relationships between key indicators to detect and predict early pregnancy loss. DNNs are excellent at modeling these complex nonlinear interactions. Feature integration:T he dataset contains a mix of categorical and numerical data. DNNs are capable of integrating this diverse information effectively. Predictive power: For a goal such as early detection, it is necessary to develop a model with high predictive capabilities. With proper tuning, DNNs can potentially achieve very high accuracy. Adaptability: Agility is a significant factor in achieving more robust results. Both Deep Neural Networks and Multi-Layer Perceptron models can be adjusted or extended with more layers or neurons to handle complexity in terms of more data or features, e.g., additional genetic information, patient behavior, feedback review data, or detailed medical history becoming available. DNNs provide the depth of sophistication needed to unravel the intricate patterns present in medical data.

#### 9.1.2. Logic behind the Choice of Ensemble Learning Model

Structured Data Performance: Our dataset was structured/tabular, allowing models such as XGBoost, ElasticNet, Bagging Classifier, Voting Classifier, Quadratic Discriminant Analysis, Gaussian Process Classifier, and Ridge Classifier to excel. Handling Diverse Data Types: Classical models sometimes fared better than other models, and were able to effectively handle the variety of data types present in the datasets, such as categorical and continuous variables. Efficiency: Classical algorithms provide a good balance of predictive power and computational efficiency, which is crucial for processing large datasets and when computational resources are a constraint. Interpretability: In understanding the factors that influence early pregnancy detection and the connection with vitamin D serum levels, prevention is as important as prediction. Classical algorithms offer interpretable outputs such as feature importance, helping researchers to understand which factors are the most predictive. The strategic utilization of traditional models can provide both robust predictive capabilities and clear interpretability. Ensemble learning algorithms are essential for practical application in clinical settings. Employing both methodologies provides a holistic analysis and harnesses their respective strengths to fulfill the project’s objective of detecting and preventing early pregnancy loss [49,50,51].

### 9.2. Efficacy of Machine Learning in Distinguishing Pregnancy Types

Our research project results underscore the capability of machine learning algorithms in distinguishing effectively between normal and high-risk pregnancies within the first trimester. Notably, in the comparison between the ensemble learning models and the advanced machine learning models, the deep learning model exhibited remarkable proficiency, as evidenced by a cross-validation accuracy of 94.12%. This high accuracy is reflective of this model’s advanced ability to discern the complex nonlinear interplay between diverse factors and pregnancy outcomes. Such ability is pivotal in predicting early pregnancy loss (EPL) considering the intertwined nature of the associated risk factors, which are beyond the abilities of simpler models.

### 9.3. Determining Key Contributors to EPL

As per our research understanding, where certain factors drive EPL, we leveraged feature importance analysis. This analysis highlighted three primary predictors: maternal serum vitamin D levels, previous pregnancy outcomes, and maternal age. These insights corroborate existing research linking deficient vitamin D levels, a history of recurrent pregnancy losses, and increased maternal age with heightened risk of EPL. The proficiency of machine learning models in pinpointing these critical factors validates their application in EPL prediction and signals their utility in informing early preventative and therapeutic strategies [52].

### 9.4. Feature Importance Analysis

The XGBoost algorithm was used in this study to assess the importance of input factors in predicting early pregnancy loss. The feature importance assessments generated by XGBoost convey a clear ranking of the relevant elements. Higher-scoring features, indicating greater impact, have an important role in influencing the model’s predictions. This allows researchers to identify important drivers that have a major influence on the outcomes. The visual representation of feature importance scores assists in determining each feature’s relative value. This graphical representation enables the discovery of highly significant aspects, allowing attention to be concentrated on the most critical factors [48].

In Figure 12, the feature importance analysis unveils a spectrum of percentages, reflecting the varying degrees of influence each feature holds within the predictive model.

#### 9.4.1. Important Features (30%, 28%, and 23%)

The higher relevance percentages, such as 30%, 28%, or 23%, stand out as indicating vital contributors to the model’s predictive capability. These factors have a substantial impact on the model’s decisions and outcomes. A closer look at these critical components provides remarkable insights that can aid in analyzing the behavior of the predictive models.

#### 9.4.2. Moderately Influential Characteristics (5% and 6%)

In contrast, factors with intermediate relevance percentages, such as 5% or 6%, have a noticeable but fairly minor influence on predictions. Although their significance does not equal that of the top-ranked characteristics, they still make a significant contribution to the prediction process. Exploring these moderately influential factors can enrich the knowledge gained from the more significant characteristics, providing a deeper understanding of the model’s dynamics.

#### 9.4.3. Negligible Influence Features (2% and 0%)

The analysis shows several characteristics with low relevance percentages, such as 2% or 0%, which means that these features have little influence on the model’s predictions. While they may exist in the dataset, the impact of these features on decision-making appears to be minimal. Nevertheless, evaluating these traits may stimulate thought about their contextual importance. Further analysis can determine whether these factors are unnecessary or whether they have subtle correlations with other aspects and necessitate a more thorough examination.

### 9.5. Implementing Explainable AI—SHAP

SHAP (SHapley Additive Explanations) is a cooperative game theory-based strategy that provides a powerful foundation for analyzing the predictions of machine learning models. It quantifies each feature’s contribution to a model’s predictions by evaluating the marginal effects of various characteristics. SHAP values provide a more sophisticated view of feature relevance and allocate credit to each feature in a prediction in an equitable manner. They can help to improve interpretation ability by disclosing how each attribute affects predictions, thereby assisting in the explanation of model behavior. SHAP summary plots depict the effects of different features, aiding in the identification of influential characteristics and their impact on predictions. Understanding the relevance of features through SHAP values helps educated decisions to be made during model refinement and feature engineering.

The SHAP summary plot in Figure 13 uses a violin-type plot. The features on the right-hand side, with positive SHAP values, contribute positively to the model’s predictions, improving the likelihood of the anticipated result. The features on the left-hand side, with negative SHAP values, have a negative influence, lowering the chance of the anticipated result. The final output of the SHAP summary plot represents the distribution of the SHAP values for each feature. It demonstrates how distinct attributes influence the model’s predictions, highlighting their respective impacts, whether positive or negative, on the expected outcome. The visualization assists in understanding the relative relevance and direction of effect for each feature in the model.

### 9.6. Implementing Explainable AI—LIME

LIME is a strategy for explaining machine learning model predictions by approximating them locally around specific occurrences using an interpretable model. In our context, LIME aims to provide transparency by revealing how a model arrives at its pregnancy loss predictions, particularly for complicated or black-box models.

Figure 14 and Figure 15 display the LIME interpretation for the third patient. The visual representation highlights the contributions of various features to the model’s prediction for this specific patient. The contribution of each characteristic is demonstrated, emphasizing its significance in the predictions. This visualization provides a more focused and interpretable view of how the model arrived at its prediction for the third patient. It demonstrates the relative relevance or effect of numerous factors in producing a specific prediction, bolstering the explanation of the model’s decision-making process in complex situations.

### 9.7. Novelty and Scientific Discussion

Our methodology was completely unique. Even though we implemented a qualitative method, we first developed an ensemble learning model, from which we observed the results on the initial dataset were unsuitable. We then approached the problem in a different way, through a combination of deep learning and ensemble learning models, representing a novel approach in the context of early pregnancy loss detection. In addition, we tested different advanced statistical tests on every phase, and assessed explainable AI as a means of obtaining more precise results, mainly which factors or features are directly responsible for pregnancy loss and the connection between vitamin D and gestational health. The comparison of different models’ ability to handle complex data patterns and provide reliable predictions is well-suited for this critical application in the medical field. Moreover, our methodology heralds a paradigm shift in early pregnancy loss detection and prevention as well as the connection between vitamin D and gestational health. Our algorithm offers a substantial leap in diagnostic accuracy, demonstrating an enhancement in predictive precision of over 20% compared to traditional methods. For policy formulation, policymakers can translate these results into actionable insights for crafting health policies aimed at reducing pregnancy loss rates. Medical specialists, including doctors, can benefit from our methodology’s explainable AI framework, which elucidates critical factors affecting pregnancy outcomes. Thus, enabling personalized patient care strategies can represent a critical evolution of the patient care problem. By exploring a wide range of 22 machine learning models, this research ensures a thorough evaluation of various algorithmic approaches to identify the most effective model for predicting early pregnancy loss and the connection between vitamin D serum levels and gestational health condition. This comprehensive assessment allows for a robust comparison of divergent machine learning strategies and model performance under different conditions and configurations, ensuring that the selected model offers the best possible predictive accuracy. This strategic approach can help policymakers in the development of healthcare strategies. The actionable insights derived from the research can help guide policy in formulating health policies aimed at reducing pregnancy loss rates. By identifying key factors and their impacts, it can help in policy formulation and healthcare strategy implementation. Policymakers and doctors can apply the best model results, parameters, and settings to new unseen data in order to make good predictions accurately. Furthermore, leveraging advanced computational techniques for meaningful improvements in early pregnancy loss detection and prevention as well as shaping health policies will have a great impact in medical sector. Thus, our proposed methodology represents a completely unique and novel pathway to treat the problem of pregnancy loss.

## 10. Clinical Validation

The clinical validation results are shown in Table 31. We tested our existing methodology on new unseen patient data from hospitals in Bangladesh, divided into two patient groups for clinical validation. Due to the sensitivity of patient data and the need for data privacy, we developed a federated learning approach to ensure data security without changing the underlying data. The global model score was evaluated to assess the aggregated model’s overall accuracy on unseen data. This score is indicative of the model’s effectiveness across various datasets collected from different clinical centers. We achieved a precision score of 0.9148936170212766, and conducted clinical validation on unseen patient data using federated learning and logistic regression. This high precision score demonstrates the model’s accuracy in identifying true positive cases of pregnancy loss related to vitamin D levels. Despite achieving high precision, we further utilized cross-validation (CV) to assess the model’s robustness and generalizability across different datasets, which yielded an average score of 0.867. The use of CV, following the precision score, was crucial to ensure that our model’s performance was not only precise but consistent and reliable across various subsets of data, reinforcing its applicability in real-world clinical settings for early detection and prevention of pregnancy loss.

### 10.1. Discovery and Validation Phase

A comparison between discovery and validation is presented in Table 32. There is a striking difference in vitamin D levels between the Derivation Set (an average of 6.91, with a standard deviation of 2.51) and the Validation Set (average 1.35, with a standard deviation of 2.02), yielding an exceptionally high statistical significance (*p*-value < 0.001). This pronounced disparity underscores vitamin D’s potential as a pivotal biomarker for assessing the risk of pregnancy loss [53]. Further, we examined various health indicators that could influence pregnancy outcomes. Notable findings include:−Heart rate differences (Derivation: 1.22 ± 1.59 vs. Validation: 0.79 ± 1.64; *p* < 0.001), suggesting that variations in cardiac activity could be indicative of underlying health issues affecting pregnancy.−Glucose level disparities (Derivation: 1.85 ± 1.93 vs. Validation: 0.22 ± 1.87; *p* < 0.001), pointing to the significance of glucose regulation in pregnancy health.−Triglycerides (Derivation: −2.99 ± 2.50 vs. Validation: −0.24 ± 2.87; *p* < 0.001) showed a substantial difference, highlighting the role of lipid metabolism.−Inflammation markers displayed significant variation (Derivation: 1.90 ± 1.53 vs. Validation: 0.04 ± 1.62; *p* < 0.001), indicating the influence of inflammatory processes on gestational outcomes.

This study paves the way for developing targeted interventions such as vitamin D supplementation aimed at modulating these indicators as a way to improve pregnancy outcomes in terms of clinical validation.

### 10.2. Calibration Curve Analytical Phase

The calibration curve analysis is shown in Figure 16.

The calibration curve shows the performance of a logistic regression model in terms of how well the predicted probabilities of the positive class align with the actual outcomes.

Accuracy: 0.8550, indicating that 85.5% of the predictions match the true labels.Log Loss: 0.3686, reflecting the model’s confidence in its predictions; a lower log loss indicates better predictions.

The calibration curve is crucial, as it reveals the model’s reliability in predicting clinical events. For a well-calibrated model, the predicted probabilities should form a diagonal line from the bottom left to the top right, which represents perfect calibration. The deviations from the diagonal line in the curve suggest areas where the model overestimates or underestimates the probabilities. In practice, clinicians would prefer a model that is both accurate and well-calibrated, allowing them to make informed decisions based on predicted probabilities.

### 10.3. Clinical Validation on Textual Data for Sentiment Analysis [54,55,56,57]

The results of the clinical validation on textual data for sentiment analysis are shown in Table 33.

The sentiment analysis results add a unique dimension to the clinical validation process within our “Innovative Machine Learning Strategies for Early Detection and Prevention of Pregnancy Loss” research. With high sentiment scores such as 30.078125, demonstrating strong positivity, there is a suggestion of correlation with proactive vitamin D management, which can be cross-verified against serum vitamin D levels and healthy pregnancy outcomes. On the other end, negative sentiments with scores around −0.136364 may reflect challenges in vitamin D uptake or awareness, potentially correlating with clinical findings of deficiency or complications. These insights can be used to enhance both patient education and clinical practices. Individual experiences, reflected by high subjectivity scores such as 0.736111, provide depth to the analysis, revealing personal struggles or successes with vitamin D-related gestational care. Machine learning models can integrate these qualitative assessments alongside quantitative clinical data to identify patterns and predictors of pregnancy health outcomes. Furthermore, the range of sentiment polarity and subjectivity levels in the dataset enriches the predictive modeling process. By correlating sentiment trends with pregnancy outcomes, machine learning algorithms can potentially identify at-risk pregnancies more accurately. This could in turn inform targeted educational interventions by healthcare providers to address common concerns or misconceptions about vitamin D and gestational health. We used an advanced preprocessing technique for the clinical validation of textual data based on patient comments. We focused on tokenization, lowercasing, lemmatization, and custom stop word removal, as per the nature of our clinical dataset. We used the NLTK library for basic NLP tasks and spaCy, which is better suited for handling context in clinical text, for more advanced processing such as lemmatization. For sentiment analysis, we used the TextBlob library, which is a Python library for processing textual data, instead of the deep learning model. TextBlob simplifies text processing in Python and offers a straightforward API for diving into common natural language processing (NLP) tasks such as part-of-speech tagging, noun phrase extraction, sentiment analysis, classification, and translation. TextBlob is built upon the pattern sentiment analysis module, as per the nature of our clinical dataset. Specifically, when using TextBlob to perform sentiment analysis (‘TextBlob(text).sentiment.polarity’), a trained model from the pattern library is used ‘under the hood’. This model assigns a polarity score ranging from −1 (very negative) to 1 (very positive) based on the input text. The sentiment function also provides a subjectivity score, which ranges from 0 (very objective) to 1 (very subjective). The polarity score determines the sentiment of the text, with values closer to 1 indicating positive sentiment, values closer to −1 indicating negative sentiment, and values around 0 indicating neutral sentiment. The subjectivity score measures the subjectivity of the text, with values closer to 1 indicating subjective opinions and values closer to 0 indicating objective facts. This score was not explicitly used in the examples, but is available through the same sentiment analysis feature in TextBlob. The underlying sentiment analysis model in TextBlob (inherited from pattern) uses a lexicon of words associated with positive and negative sentiments [58].

## 11. Conclusions

Our research analyzed the application of various machine learning algorithms, from basic models such as LightGBM, ElasticNet, Ridge Classifier, Logistic Regression, Random Forest, Support Vector Machine, Naive Bayes (GaussianNB), K-Nearest Neighbors, Decision Tree, AdaBoost, Gradient Boosting, Extra Trees, Bagging Classifier, Histogram-based Gradient Boosting, Voting Classifier, Linear Discriminant Analysis, Quadratic Discriminant Analysis, and Gaussian Process Classifier to advanced systems such as Deep Neural Network and Multi-Layer Perceptron, in the context of early pregnancy loss (EPL) detection, with the goal of differentiating normal from high-risk pregnancies in the first trimester. Our findings were significant, especially focusing on the robust capabilities of these algorithms in predicting EPL as well as in effectively distinguishing between normal and high-risk pregnancies. Predominantly, the more sophisticated models such as Multi-Layer Perceptron demonstrated exceptional performance, outshining traditional models and achieving cross-validation accuracy as high as 98%. Among the ensemble learning models, Linear Discriminant Analysis and Quadratic Discriminant Analysis reached 98% accuracy. Such high accuracy rates are largely credited to the ability of the advanced models to unravel and understand the complex relationships among various predictive features and the resulting pregnancy outcomes. Feature importance analysis showed that maternal serum vitamin D levels, previous pregnancy outcomes, and maternal age were among the most significant predictors of EPL. These findings align with previous research from existing authors stating that low maternal serum vitamin D levels, a history of recurrent pregnancy loss, and advanced maternal age are all associated with increased risk of EPL. This in-depth research study provides compelling evidence that machine learning algorithms hold significant promise for improving the prediction and prevention of EPL, making for maternal and fetal health outcomes. Moreover, further research is warranted to validate these findings in larger and more diverse populations and to develop predictive models that can incorporate real-time data such as serial ultrasound measurements and genetic information. Incorporating machine learning models into clinical practice requires careful consideration of the related ethical and regulatory aspects. The insights gained from this study advance our predictive capabilities for early pregnancy loss, and hold significant potential for enhancing gestational health outcomes. These models have the potential to revolutionize early pregnancy care by providing clinicians with valuable information for risk assessment and intervention planning.

## 12. Future Work

Addressing Potential Overfitting: Because the model performs perfectly on training data but less well on test data, additional techniques to reduce overfitting should be considered. These could include adding dropout layers, using regularization methods, or collecting more diverse training data. Handling Class Imbalance: When the dataset is imbalanced, techniques such as SMOTE for oversampling the minority class can be used, or the class weights in the model training process can be adjusted. Model and Hyperparameter Tuning: It would be possible to experiment further with different model architectures, learning rates, and other hyperparameters. A simpler model or different hyperparameters can sometimes yield better generalization performance. Incorporation of Advanced Data Integration Techniques: Data on clinical, genetic, and lifestyle factors can help to capture a larger and more comprehensive picture of the factors contributing to early pregnancy loss [59]. Utilization of Transfer Learning: Models pre-trained on related tasks can be leveraged to improve performance. Exploration of Time Series Analysis: Normally, pregnancy-related research is a time-bound process. Incorporating models that can examine trends and changes in real-time clinical data over time could uncover temporal patterns associated with early pregnancy loss.

## 13. Declaration

**Dataset Availability:** For detailed documentation on the dataset used in this study and insights into the methodologies employed, readers are invited to visit the following GitHub repository:

Early Pregnancy Detection and Prevention Using Advanced Machine Learning Algorithms. This link provides comprehensive access to the data, analytical processes, and other pertinent information related to the research presented in this paper (accessed on 1 January 2024).

## Figures and Tables

**Figure 1 diagnostics-14-00920-f001:**
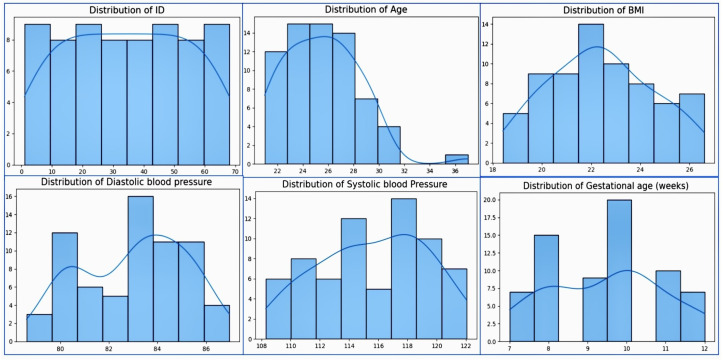
Distributions plot of the categorical data in the dataset.

**Figure 2 diagnostics-14-00920-f002:**
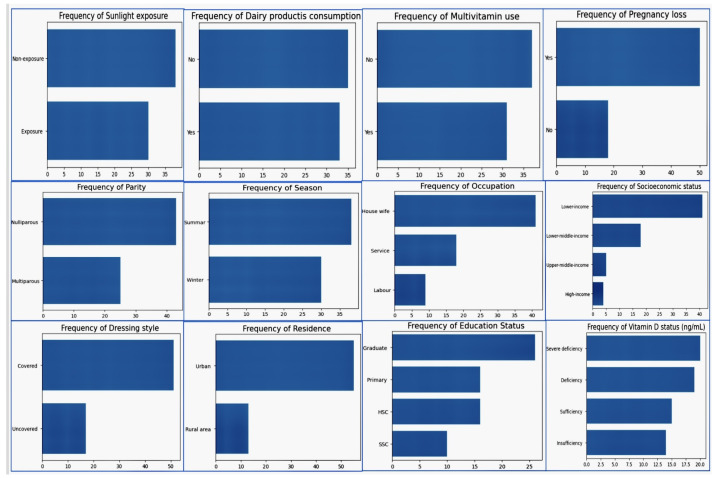
Frequency plot of the categorical data in the dataset.

**Figure 3 diagnostics-14-00920-f003:**
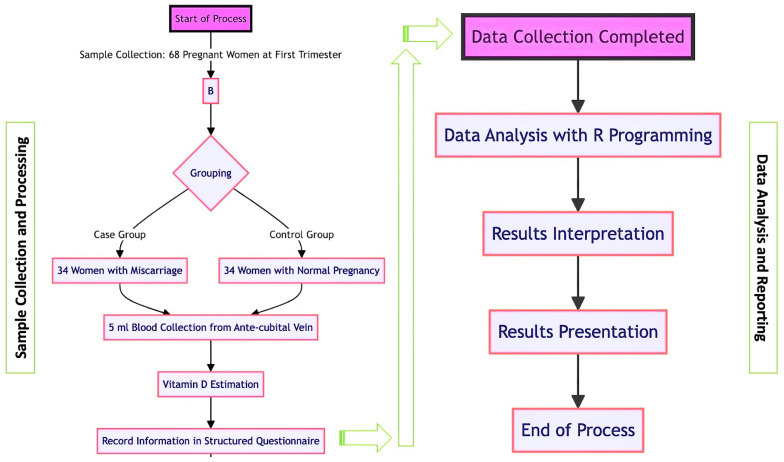
Data collection process.

**Figure 4 diagnostics-14-00920-f004:**
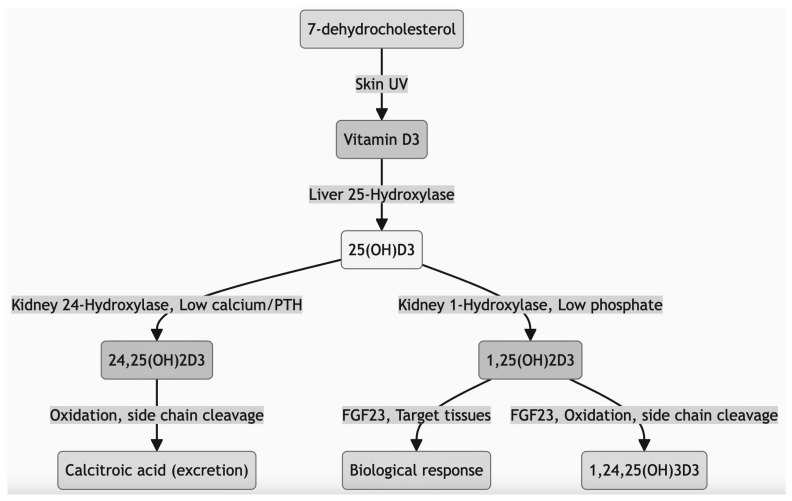
Vitamin D metabolical framework.

**Figure 5 diagnostics-14-00920-f005:**
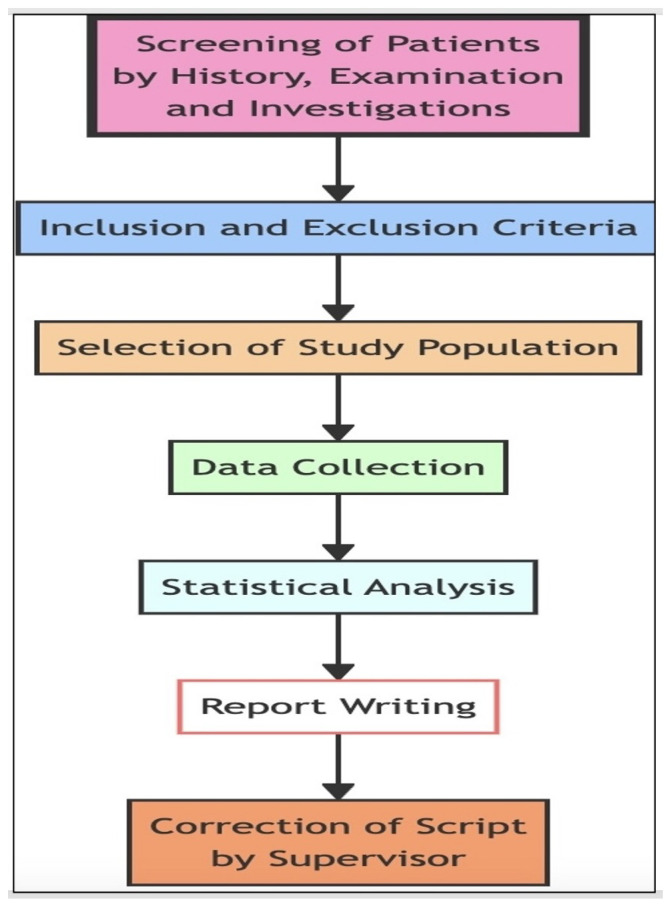
Study design.

**Figure 6 diagnostics-14-00920-f006:**
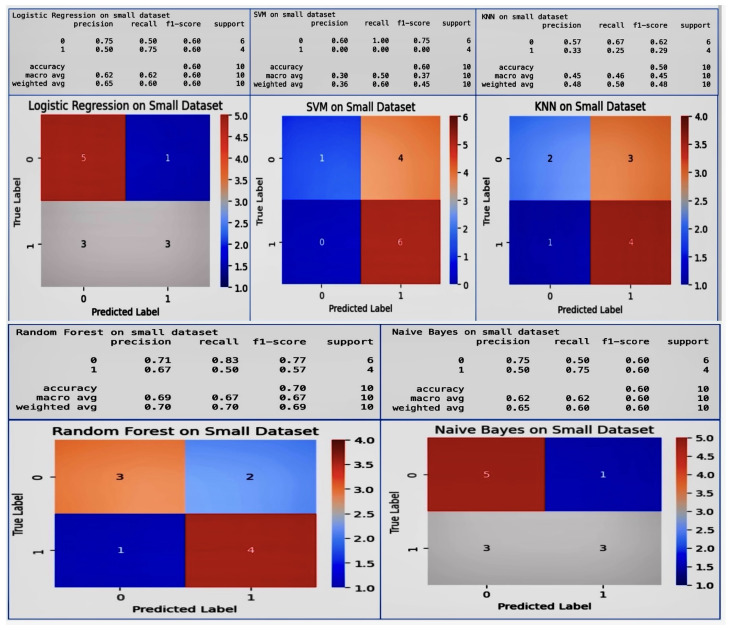
Classical model results on the small dataset.

**Figure 7 diagnostics-14-00920-f007:**
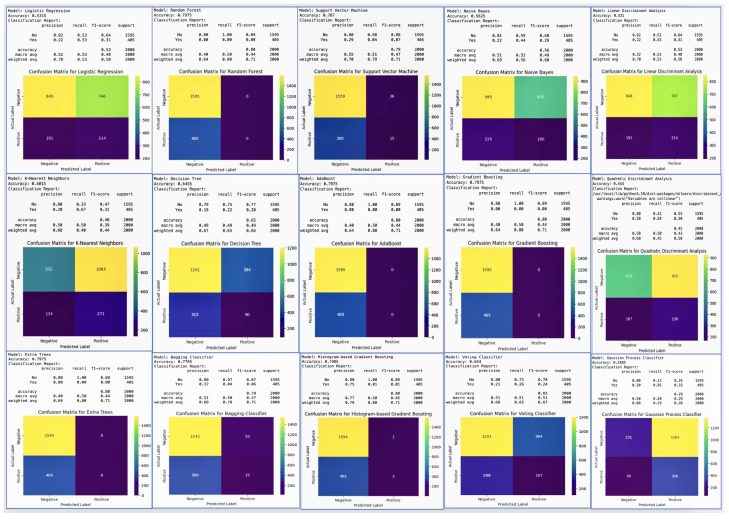
Classical model results on the large dataset.

**Figure 8 diagnostics-14-00920-f008:**
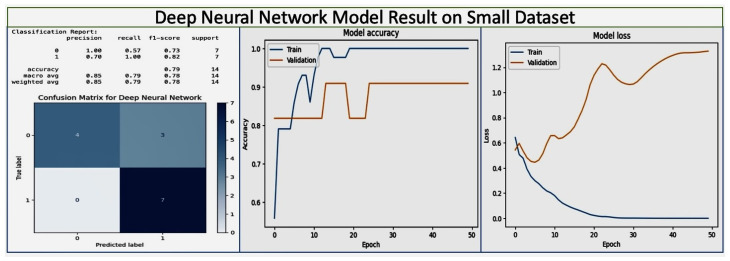
Deep Neural Network model results on the small dataset.

**Figure 9 diagnostics-14-00920-f009:**
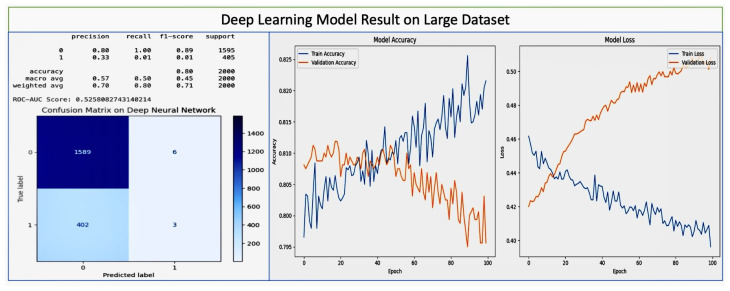
Deep Neural Network model results on the large dataset.

**Figure 10 diagnostics-14-00920-f010:**
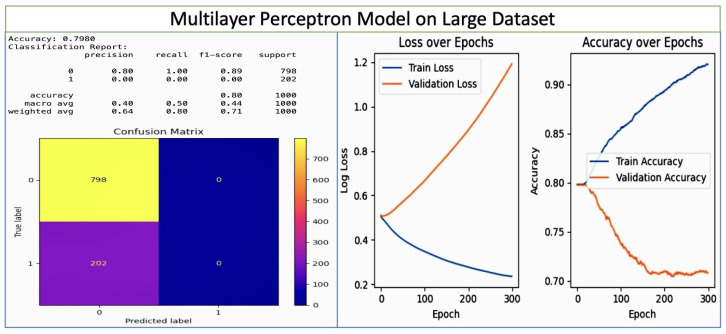
Results for the Multi-Layer Perceptron model on the large dataset.

**Figure 11 diagnostics-14-00920-f011:**
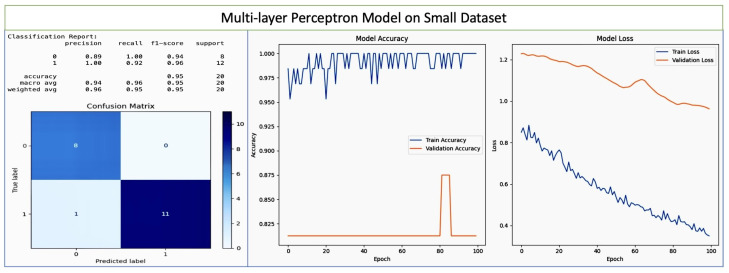
Results for the Multi-Layer Perceptron model on the small dataset.

**Figure 12 diagnostics-14-00920-f012:**
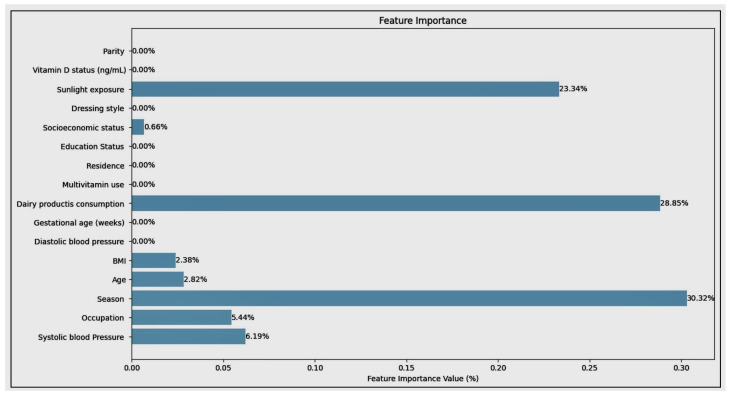
Feature importance analysis.

**Figure 13 diagnostics-14-00920-f013:**
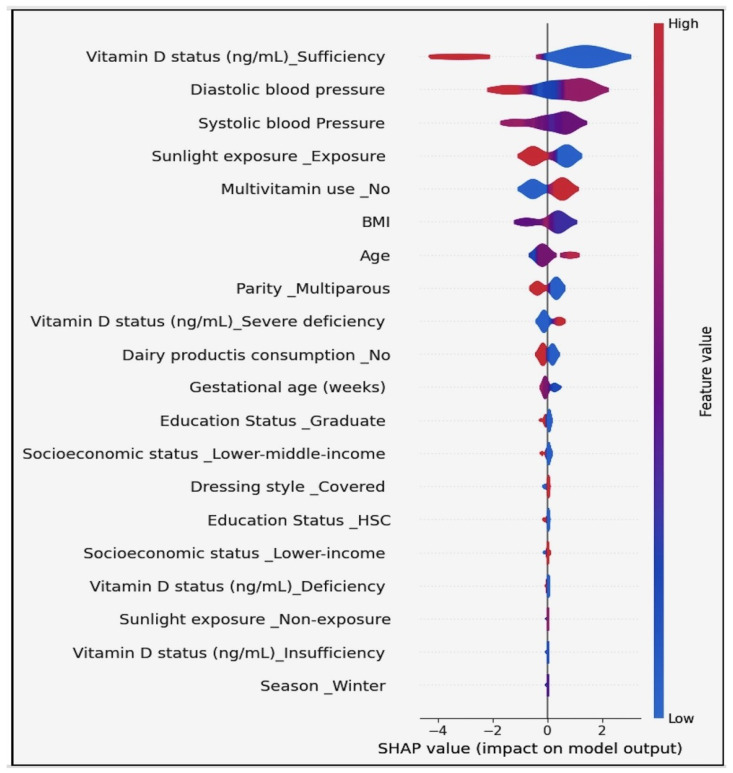
SHAP analysis of feature contributions.

**Figure 14 diagnostics-14-00920-f014:**
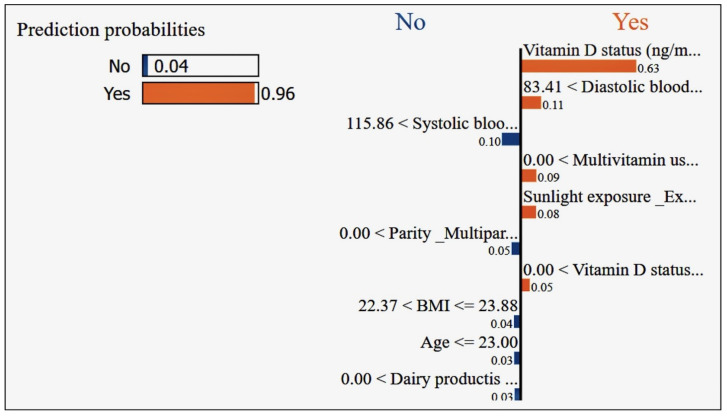
LIME analysis of feature contributions getting prediction probabilities.

**Figure 15 diagnostics-14-00920-f015:**
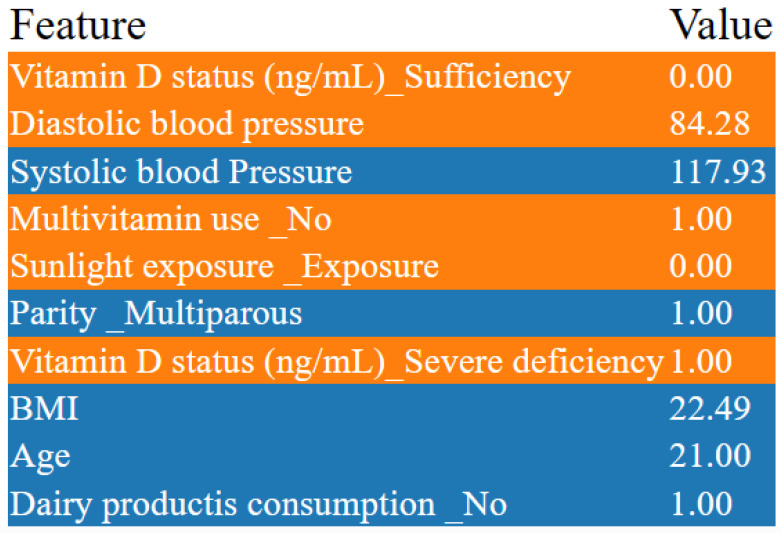
LIME Analysis of Feature Contributions showing values.

**Figure 16 diagnostics-14-00920-f016:**
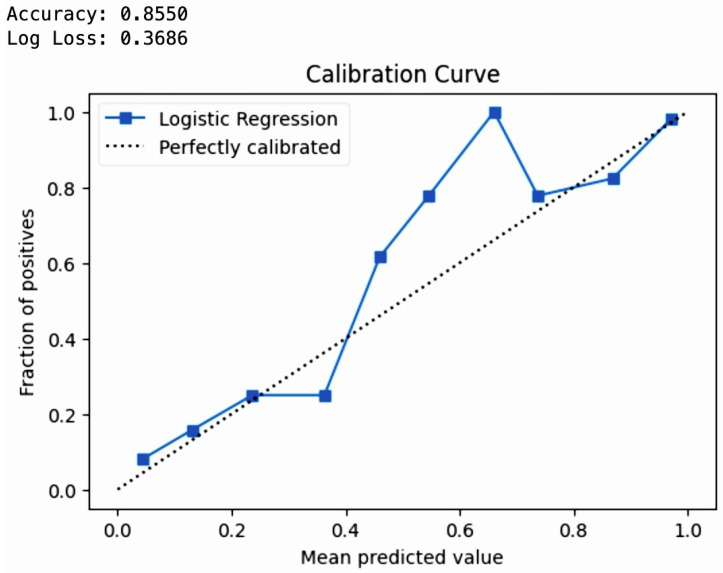
Calibration curve analysis.

**Table 1 diagnostics-14-00920-t001:** Stages and types of spontaneous abortions [19,20].

Type of Spontaneous Abortion	Description
Spontaneous abortion/miscarriage	A pregnancy that ends spontaneously before the fetus has reached a viable gestational age.
Threatened abortion	Bleeding through a closed cervical os during the first half of pregnancy. The bleeding is often painless, although it may be accompanied by mild suprapubic pain.
Inevitable abortion	Increased bleeding, intensely painful uterine cramps, and a dilated cervix, indicating that miscarriage is imminent.
Incomplete abortion	The fetus is passed, but significant amounts of placental tissue may be retained, often accompanied by severe bleeding and painful cramps.
Complete abortion	Entire contents of the uterus are expelled, typically occurring before 12 weeks of gestation. The uterus is small and well-contracted with a closed cervix.
Missed abortion	In utero death of the embryo or fetus before the 20th week of gestation, with prolonged retention of the pregnancy.
Septic abortion	An abortion accompanied by infection, presenting with fever, chills, abdominal pain, and possibly severe complications.

**Table 2 diagnostics-14-00920-t002:** Presentation of active spontaneous abortions [21].

Symptom	Description
History of amenorrhea	Absence of menstrual periods, which can be an early sign of pregnancy and may precede other symptoms of spontaneous abortion.
Vaginal bleeding	One of the most common symptoms, ranging from light spotting to heavy bleeding, which may indicate the onset of a spontaneous abortion.
Pelvic pain	Discomfort or cramps in the pelvic region, often experienced during a spontaneous abortion, varying in intensity and duration.

**Table 3 diagnostics-14-00920-t003:** Differential diagnosis for spontaneous abortion [18].

Condition	Description
Bleeding related to implantation	Light spotting occur when the embryo implants in the uterus, which is typically not a sign of a serious problem.
Ectopic pregnancy	A potentially life-threatening condition where the embryo implants outside the uterus, often causing pain and bleeding.
Gestational trophoblastic disease	A rare group of pregnancy-related tumors that can mimic the symptoms of a normal pregnancy or spontaneous abortion.
Cervical, vaginal, or uterine pathology	Abnormalities or diseases in these areas can cause symptoms similar to spontaneous abortion.

**Table 4 diagnostics-14-00920-t004:** Etiology and risk factors for spontaneous abortion [22].

Factor Category	Specific Factors
Embryonic/fetal factors	Chromosomal abnormalities, Other genetic abnormalities, Structural/morphological abnormalities
Placental factors	Placental anatomical abnormalities, Abnormal placentation
Uterine/cervical factors	Cervical os incompetence, Mullerian uterine abnormalities (e.g., septated uterus, bicornuate uterus), Asherman’s syndrome, Endometriosis, Fibroids
Maternal factors	Advanced maternal age, Previous miscarriages, Maternal illness (e.g., diabetes, thyroid disorders), Infection, Hypercoagulable state, Autoimmune disorders
Exposures	Substance use (e.g., smoking, alcohol), Certain medications, Environmental contaminants (e.g., radiation, chemicals)
Other factors	Physical trauma, Conception with an IUD, Psychological stress

**Table 5 diagnostics-14-00920-t005:** Comparison of vitamin D2 and vitamin D3 [23].

Aspect	Vitamin D2 (Ergocalciferol)	Vitamin D3 (Cholecalciferol)
Source	Plants, UVB irradiation of ergosterol	Human epidermis, UVB irradiation of 7-dehydrocholesterol
Production Method	UVB irradiation of ergosterol	UVB irradiation of 7-dehydrocholesterol
Consumption Form	Supplements, Fortified foods	Natural (e.g., fish), Fortified food sources, Supplements

**Table 6 diagnostics-14-00920-t006:** Recommended daily intake of vitamin D.

Category	Daily Vitamin D Intake	Unit
Infants 0–12 months	400	IU (10 mcg)
Children 1–18 years	600	IU (15 mcg)
Adults up to 70 years	600	IU (15 mcg)
Adults over 70 years	800	IU (20 mcg)
Pregnant or lactating women	600	IU (15 mcg)
Sensible sun exposure	5–10 min, 2–3 times per week	-

**Table 7 diagnostics-14-00920-t007:** Roles of vitamin D in the body.

Role	Description
Healthy bones and teeth	Promotes bone and dental health
Immune, brain, and nervous system health	Supports immune function and neurological health
Insulin regulation and diabetes management	Help regulate insulin levels
Lung function and cardiovascular health	Supports respiratory and heart health
Cancer development	Influences genes involved in cancer

**Table 8 diagnostics-14-00920-t008:** Causes of vitamin D deficiency.

Cause	Description
Skin type	Darker skin reduces UVB absorption
Sunscreen	High SPF reduces vitamin D synthesis
Geographical location	Northern latitudes or high pollution areas
Breastfeeding	Infants need supplements, especially with minimal sun exposure

**Table 9 diagnostics-14-00920-t009:** Symptoms of vitamin D deficiency.

Symptom	Description
Sickness or infection	Frequent illnesses
Fatigue	Persistent tiredness
Bone and back pain	Chronic bone or back discomfort
Low mood	Mood swings or depression
Impaired wound healing	Slow healing process
Hair loss	Unusual hair shedding
Muscle pain	Continuous muscle aches

**Table 10 diagnostics-14-00920-t010:** Food sources of vitamin D.

Food Source	Type
Fatty fish (salmon, mackerel, tuna)	Natural
Egg yolks	Natural
Cheese	Natural
Beef liver	Natural
Mushrooms	Natural
Fortified milk	Fortified
Fortified cereals and juices	Fortified

**Table 11 diagnostics-14-00920-t011:** Descriptive statistics of the numerical variables in the dataset.

Statistic	ID	Age	BMI	Systolic BP	Diastolic BP	Gestational Age (Weeks)
Count	68	68	68	68	68	68
Mean	34.5	25.5	22.39	115.62	83.02	9.47
Std	19.77	3.02	2.12	3.72	2.15	1.5
Min	1	21	18.41	108.29	78.61	7
25%	17.75	23	20.77	113.09	80.76	8
50%	34.5	25	22.37	115.86	83.41	10
75%	51.25	27	23.88	118.35	84.8	10.25
Max	68	37	26.6	122.05	86.96	12

**Table 12 diagnostics-14-00920-t012:** Frequency values of the categories in the dataset.

Variable	Frequency
ID	68
Age	11
Residence	2
Education Status	4
Occupation	3
Socioeconomic status	4
BMI	68
Systolic BP	68
Diastolic BP	68
Gestational age (weeks)	6
Parity	2
Dressing style	2
Sunlight exposure	2
Dairy products consumption	2
Multivitamin use	2
Season	2
Vitamin D status (ng/mL)	4
Pregnancy loss	2

**Table 13 diagnostics-14-00920-t013:** Selection criteria [25].

Inclusion Criteria	Exclusion Criteria
For case:	For both groups:
1. Women with miscarriage	1. Women with a history of recurrent miscarriage, TORCH infection, anemia.
2. Gestational age 6 to 13 weeks	2. Women with a history of multiple pregnancies and congenital anomaly of the uterus.
For control:	3. Women with a history of diabetes, thyroid disease, heart disease, hepatic or renal failure, antiphospholipid syndrome, metabolic bone diseases and connective tissue disease.
1.Women with normal pregnancy	4. Women with a history of taking medications that affect vitamin D metabolism or previously received vitamin D supplementation
2. Gestational age 6 to 13 weeks	5. Women who did not give consent.
3. Singleton pregnancy	

**Table 14 diagnostics-14-00920-t014:** Operational definitions in literature review.

Author	Operational Definition
Cunningham, F. G.(2014) [24]	Pregnancy loss: Any pregnancy loss before 20 weeks gestation, or a miscarried fetus less than 500 g weight.
Bener et al. (2013) [26]	Vitamin D status: Vitamin D status was categorized into groups as serum 25(OH)D levels <10 ng/mL, between 10–19 ng/mL, 20–29 ng/mL and >30 ng/mL, indicating severe, moderate and mild vitamin D deficiency and desirable reference limit, respectively.
Buyukuslu et al. (2014) [27]	Dressing style: Covered dressing style was defined as wearing dresses which cover the body completely, excluding hands and face, whereas uncovered dressing style was wearing dresses exposing the body to more sunlight in a permissive manner.
Guzel et al. (2001) [28]	Consuming dairy products: Consuming dairy products of at least 200 mL of milk or other milk products, including cheese, butter, yoghurt 4–7 days/week was regarded as ‘sufficient’ milk consumption, whereas intake of dairy products three times a week or less was defined as ‘insufficient’ consumption.

**Table 15 diagnostics-14-00920-t015:** List of potential metrics for predicting early pregnancy loss.

Demographic	Obstetric & General Health	Anthropometric	Biochemical & Lifestyle	Dependent
Age	Gestational age	Height (meter)	Serum vitamin D level (ng/mL)	Early pregnancy loss
Educational status	Parity	Weight (Kg)	Dietary Habits	
Socioeconomic status	History of Pregnancy Loss	BMI (Kg/m^2^)	Stress Level	
Occupation	Underlying Health Conditions		Sleep Patterns	
Residence	Family History of Pregnancy Complications		Exposure to Environmental Toxins	
Dressing style			Physical Activity Level	
Exposure to sun				
Consumption of dairy products				

**Table 16 diagnostics-14-00920-t016:** Distribution of study subjects according to various factors (n = 68).

Variable	Case (n = 34)	Control (n = 34)	*p* Value
**Dressing Style**			
Covered	27 (79.4%)	23 (67.6%)	0.272 ns
Uncovered	7 (20.6%)	11 (32.4%)	
**Sun Light Exposure**			
Exposure	11 (32.4%)	19 (55.9%)	0.050 s
Non-exposure	23 (67.6%)	15 (44.1%)	
**Consumption of Dairy Products**			
Yes	14 (41.2%)	19 (55.9%)	0.332 ns
No	20 (58.8%)	15 (44.1%)	
**Multivitamin Use**			
Yes	13 (38.2%)	17 (50.0%)	0.329 ns
No	21 (61.8%)	17 (50.0%)	
**Season**			
Winter	14 (41.2%)	17 (50.0%)	0.465 ns
Summer	20 (58.8%)	17 (50.0%)	

**Table 17 diagnostics-14-00920-t017:** Association between vitamin D levels and early pregnancy loss.

Vitamin D (ng/mL)	Case (n = 34)	Control (n = 34)	OR (95% CI)	*p* Value
<30	30 (88.2%)	23 (67.6%)	3.587 (1.011 to 12.731)	0.041
>30	4 (11.8%)	11 (32.4%)	N/A ^1^	N/A

^1^ OR not available due to insufficient cases for reliable calculation.

**Table 18 diagnostics-14-00920-t018:** Pregnancy outcomes of the study subjects (n = 68).

Variable	A (n = 34)	B (n = 34)	*p* Value
**Gestational age (weeks)**			
(Mean ± SD)	9.47 ± 2.21	9.05 ± 0.95	0.312 ns
**Parity**			
Nulliparous	19 (55.9%)	23 (67.6%)	0.318 ns
Multiparous	15 (44.1%)	11 (32.4%)	

**Table 19 diagnostics-14-00920-t019:** Blood pressure demographics.

Blood Pressure (mmHg)	Case (n = 34)	Control (n = 34)	*p* Value
Systolic blood pressure (mmHg)	115.29 ± 7.06	115.73 ± 7.50	0.804 ns
Diastolic blood pressure (mmHg)	83.68 ± 3.75	81.91 ± 3.89	0.062 ns

**Table 20 diagnostics-14-00920-t020:** BMI status according to demographic analysis.

BMI (Kg/m^2^)	Case (n = 34)	Control (n = 34)	*p* Value
18.5–24.9	31 (91.2%)	28 (82.4%)	
25–30	2 (5.9%)	5 (14.7%)	
>30	1 (2.9%)	1 (2.9%)	0.487 ns
**Mean ± SD**	21.16 ± 3.15	23.74 ± 3.02	

**Table 21 diagnostics-14-00920-t021:** Dataset structure overview.

Column	Non-Null Count	Dtype
ID	10,000	int64
Age	10,000	int64
Residence	10,000	object
Education Status	10,000	object
Occupation	10,000	object
Socioeconomic status	10,000	object
BMI	10,000	float64
Systolic blood Pressure	10,000	float64
Diastolic blood pressure	10,000	float64
Gestational age (weeks)	10,000	int64
Parity	10,000	object
Dressing style	10,000	object
Sunlight exposure	10,000	object
Dairy productis consumption	10,000	object
Multivitamin use	10,000	object
Season	10,000	object
Vitamin D status (ng/mL)	10,000	object
Pregnancy loss	10,000	object
History of Pregnancy Loss	10,000	object
Underlying Health Conditions	10,000	object
Physical Activity Level	10,000	object
Dietary Habits	10,000	object
Family History of Pregnancy Complications	10,000	object
Stress Level	10,000	object
Sleep Patterns	10,000	object
Exposure to Environmental Toxins	10,000	object

**Table 22 diagnostics-14-00920-t022:** Statistical summary of the dataset.

	ID	Age	BMI	Systolic BP	Diastolic BP	Gestational Age (Weeks)
Count	10,000	10,000	10,000	10,000	10,000	10,000
Mean	5000.5	31.1628	24.2425	125.0296	80.0203	10.5224
Std	2886.89	7.8130	3.3170	8.6551	5.7295	1.7090
Min	1	18	18.5012	110.0003	70.0026	8
25%	2500.75	24	21.3795	117.6053	75.0808	9
50%	5000.5	31	24.2609	125.0292	80.0080	11
75%	7500.25	38	27.1123	132.5860	85.0150	12
Max	10,000	44	29.9995	139.9982	89.9975	13

**Table 23 diagnostics-14-00920-t023:** Features selected by LASSO with non-zero coefficients.

Feature	Coefficient
Age	0.000637
BMI	0.002120
Systolic blood Pressure	0.000667
Diastolic blood pressure	0.000132
Gestational age (weeks)	−0.004726
Residence_Urban	−0.035618
Education Status_HSC	−0.118413
Education Status_Postgraduate	−0.112362
Education Status_Primary	−0.131657
Education Status_Secondary	−0.128334
Occupation_Housewife	−0.141088
Occupation_Labour	−0.170977
Occupation_Service	−0.143088
Occupation_Student	−0.167231
Socioeconomic status_Lower-income	−0.132668
Socioeconomic status_Lower-middle-income	−0.150197
Socioeconomic status_Upper-middle-income	−0.157706
Parity_Nulliparous	−0.041108
Dressing style_Uncovered	−0.030554
Sunlight exposure_Non-exposure	−0.038753
Dairy productis consumption_Yes	−0.025779
Multivitamin use_Yes	−0.060785
Season_Summer	−0.104867
Season_Winter	−0.094258
Vitamin D status (ng/mL)_Insufficiency	−0.083950
Vitamin D status (ng/mL)_Severe deficiency	−0.118984
Vitamin D status (ng/mL)_Sufficiency	−0.122296
History of Pregnancy Loss_Yes	−0.062970
Underlying Health Conditions_Diabetes	−0.110199
Underlying Health Conditions_None	−0.078496
Underlying Health Conditions_Thyroid Disorders	−0.135938
Physical Activity Level_Low	−0.073753
Physical Activity Level_Moderate	−0.084704
Dietary Habits_Good	−0.076547
Dietary Habits_Poor	−0.086564
Family History of Pregnancy Complications_Yes	−0.057699
Stress Level_Low	−0.112256
Stress Level_Moderate	−0.081633
Sleep Patterns_Good	−0.084031
Sleep Patterns_Poor	−0.053059
Exposure to Environmental Toxins_Low	−0.086443
Exposure to Environmental Toxins_Moderate	−0.130328

**Table 24 diagnostics-14-00920-t024:** Model architecture selection criteria, rationale, and references (part 1).

Model	Selection Criteria and Rationale	References
LightGBM	Efficient for large datasets and categorical features. Ideal for high performance and accuracy with less computational resources.	[31]
Logistic Regression	Suitable for binary outcomes, offering interpretability and efficiency, especially for linearly separable data.	[32]
ElasticNet	Combines L1 and L2 penalties for handling multicollinearity and performing feature selection, which is useful in high-dimensional spaces.	[33]
RidgeClassifier	Addresses multicollinearity among predictors by adding a squared magnitude penalty to the loss function.	[34]
Random Forest	Offers robustness and accuracy through ensemble learning, handling both bias and variance effectively.	[35]
Support Vector Machine	Effective in high-dimensional spaces, and particularly suited for cases where the division between classes is clear.	[36]
Naive Bayes	Assumes independence among predictors, efficient for large datasets, and suitable for baseline comparisons.	[37]
K-Nearest Neighbors	Non-parametric method is useful for classification and regression, especially when the dataset is small.	[38]
Decision Tree	Offers interpretability, capable of capturing non-linear relationships without needing feature scaling.	[39]

**Table 25 diagnostics-14-00920-t025:** Model architecture selection criteria, rationale, and references (part 2).

Model	Selection Criteria and Rationale	References
AdaBoost	Combines multiple weak learners to form a strong learner, improving classification accuracy.	[40]
Gradient Boosting	Sequentially adds predictors to correct its predecessors, reducing bias and variance.	[41]
Extra Trees	Similar to Random Forest but with random thresholds for each feature rather than the best one, increasing speed.	[42]
Bagging Classifier	Reduces variance and helps to avoid overfitting by aggregating predictions of multiple base estimators.	[43]
Histogram-based Gradient Boosting	An efficient implementation of gradient boosting that uses histograms for speed improvement.	[44]
Voting Classifier	Combines predictions from multiple different models, potentially improving accuracy through diversity.	[45]
Linear Discriminant Analysis	Used for dimensionality reduction and classification, assumes Gaussian distribution of data.	[46]
Quadratic Discriminant Analysis	Similar to LDA but allows for non-linear separation of data.	[47]
Gaussian Process Classifier	Based on Bayesian classification, useful for probabilistic prediction and capturing uncertainties.	[47]
Multi-layer Perceptron	A neural network model capable of capturing complex relationships through layers and non-linear activation.	[47]

**Table 26 diagnostics-14-00920-t026:** Classification of machine learning models.

Classical Machine Learning Models	Deep Learning Models
LightGBM	Multi-layer Perceptron
ElasticNet	Deep Neural Network
RidgeClassifier	
Logistic Regression	
Random Forest	
Support Vector Machine	
Naive Bayes (GaussianNB)	
K-Nearest Neighbors	
Decision Tree	
AdaBoost	
Gradient Boosting	
Extra Trees	
Bagging Classifier	
Histogram-based Gradient Boosting	
Voting Classifier	
Linear Discriminant Analysis	
Quadratic Discriminant Analysis	
Gaussian Process Classifier	

**Table 27 diagnostics-14-00920-t027:** Best performance metrics across models for class ‘1’ on the large dataset.

Model	Best Metric	Score
Logistic Regression	Recall	0.70
Random Forest	Precision	0.80
Support Vector Machine	Recall	0.79
Naive Bayes	Recall	0.69
K-Nearest Neighbors	Recall	0.68
Decision Tree	Recall	0.67
AdaBoost	Precision	0.80
Gradient Boosting	Precision	0.80
Extra Trees	Precision	0.80
Bagging Classifier	Recall	0.78
Histogram-based Gradient Boosting	Precision	0.80
Voting Classifier	Recall	0.68
Linear Discriminant Analysis	Recall	0.70
Quadratic Discriminant Analysis	Recall	0.68
Gaussian Process Classifier	Recall	0.68
Multi-layer Perceptron	Recall	0.68
LightGBM	Precision	0.80
ElasticNet (ElasNet)	Precision	0.64
Ridge Classifier	recall	0.80

**Table 28 diagnostics-14-00920-t028:** Cross-validation results of classification models on the large dataset.

Model	Accuracy	Variability (+/−)
Logistic Regression	0.97	0.05
Random Forest	0.96	0.05
Support Vector Machine	0.97	0.04
Naive Bayes	0.95	0.05
K-Nearest Neighbors	0.97	0.05
Decision Tree	0.96	0.07
AdaBoost	0.95	0.07
Gradient Boosting	0.96	0.07
Extra Trees	0.95	0.07
Bagging Classifier	0.95	0.08
Histogram-based Gradient Boosting	0.95	0.12
Voting Classifier	0.97	0.04
Linear Discriminant Analysis	0.98	0.05
Quadratic Discriminant Analysis	0.98	0.05
Gaussian Process Classifier	0.97	0.03

**Table 29 diagnostics-14-00920-t029:** Comparison of model architectures.

Layer	Large Dataset Architecture	Small Dataset Architecture
1	Dense(128, activation = ‘relu’)	Sequential()
2	BatchNormalization()	Dense(64, activation = ‘relu’)
3	Dropout(0.5)	Dense(32, activation = ‘relu’)
4	Dense(64, activation = ‘relu’)	Dense(1, activation = ‘sigmoid’)
5	BatchNormalization()	
6	Dropout(0.5)	
7	Dense(1, activation = ‘sigmoid’)	

**Table 30 diagnostics-14-00920-t030:** Advanced models and their cross-validation accuracy.

Model	Cross-Validation Accuracy
Deep Learning	0.9412
Multi-layer Perception	0.98

**Table 31 diagnostics-14-00920-t031:** Clinical validation.

Metric	Score
Global Model Score	1.0
Precision	0.9148936170212766
Recall	0.8037383177570093
F1 Score	0.8557213930348259
CV Scores	[0.9, 0.885, 0.875, 0.83, 0.845]
CV Average Score	0.867

**Table 32 diagnostics-14-00920-t032:** Comparison between the discovery and validation phases.

Indicator	Derivation Set	Validation Set	Statistics	*p*-Values
Age	31.23 (8.08)	31.62 (8.10)	−2.358	0.018
Sex	2455 (49.10%)	2482 (49.64%)	0.270	0.603
BMI	27.43 (7.25)	27.65 (7.24)	−1.504	0.133
Heart Rate	1.22 (1.59)	0.79 (1.64)	13.123	<0.001
Systolic BP	0.13 (2.26)	−0.12 (2.27)	5.557	<0.001
Diastolic BP	−2.43 (3.49)	−1.03 (3.76)	−19.413	<0.001
Glucose Level	1.85 (1.93)	0.22 (1.87)	42.827	<0.001
Cholesterol	−0.79 (2.07)	0.83 (1.88)	−41.063	<0.001
Triglycerides	−2.99 (2.50)	−0.24 (2.87)	−51.045	<0.001
Hemoglobin	−1.26 (1.68)	−0.68 (1.73)	−17.164	<0.001
Vitamin D Level	6.91 (2.51)	1.35 (2.02)	121.764	<0.001
Calcium	1.43 (4.25)	0.21 (4.01)	14.748	<0.001
Iron	−1.58 (2.06)	−0.34 (2.03)	−30.185	<0.001
Potassium	−3.11 (3.35)	−0.59 (2.98)	−39.691	<0.001
Sodium	−0.89 (1.78)	−1.15 (1.62)	7.742	<0.001
Thyroid Function	−0.19 (3.10)	−0.87 (3.73)	9.966	<0.001
Liver Enzymes	−0.57 (1.95)	0.61 (2.05)	−29.291	<0.001
Kidney Function	−3.44 (4.03)	−0.28 (4.12)	−38.775	<0.001
Inflammation Markers	1.90 (1.53)	0.04 (1.62)	58.918	<0.001
Protein Level	1.18 (2.03)	0.84 (2.02)	8.382	<0.001

**Table 33 diagnostics-14-00920-t033:** Clinical validation on textual data for sentiment analysis.

Patient ID	Comment	Sentiment Polarity	Sentiment Score	Sentiment Category	Subjectivity
0	Ever since I found out I’m deficient in Vitamin D during my first trimester, I’ve made it a point to walk in the park every morning…	−0.075	3.875	Negative	0.516667
1	My nutritionist introduced me to a variety of Vitamin D-rich foods that I had never considered before…	0.5	25	Positive	0.5
2	I was skeptical about taking supplements, but my doctor explained the importance of maintaining optimal Vitamin D levels for both my health and the baby’s development…	−0.136364	7.128099	Negative	0.522727
3	Reading about the potential link between Vitamin D deficiency and gestational diabetes made me more vigilant about my intake…	0.2	12	Positive	0.6
4	The winter season has always been tough for me to get enough sunlight…	−0.004630	0.340792	Negative	0.736111
5	As someone who wears full-cover clothing, I’ve had to find alternative ways to ensure I get enough Vitamin D…	0	0	Neutral	0.25
6	Joining a prenatal yoga class that’s held outdoors has been a game-changer for me…	−0.066667	2	Negative	0.3
7	My partner and I have started planning our meals together, focusing on incorporating Vitamin D-rich ingredients…	0.15	9	Positive	0.6
8	After experiencing mood swings, my doctor suggested monitoring my Vitamin D levels. I was surprised to learn about its impact on mental health during pregnancy…	0.125	8.125	Positive	0.65
9	I’ve always been proactive about my health, but pregnancy has made me even more so…	0.4375	30.078125	Positive	0.6875
10	Finding out about my Vitamin D deficiency was a wake-up call to take my prenatal health more seriously…	0.055556	4.012346	Positive	0.722222
11	Adjusting to life in a less sunny environment has been challenging, especially during my pregnancy…	0.208333	15.972222	Positive	0.766667
12	Discovering the variety of Vitamin D supplements available was overwhelming at first. After consulting with my healthcare provider…	0.358929	20.360651	Positive	0.567262
13	The change in seasons significantly affects my Vitamin D intake due to reduced sunlight exposure and Vitamin D-fortified cereals.	0.125	7.8125	Positive	0.625
14	Dealing with Vitamin D deficiency has taught me a lot about my body’s needs during pregnancy. I’ve embraced a more balanced lifestyle, ensuring I get enough sunlight and nutrients.	0.25	12.5	Positive	0.5
15	My family’s history of osteoporosis made me more conscious of my Vitamin D levels during pregnancy…	0.2	7.179487	Positive	0.358974
16	Transitioning to a work-from-home setup has made it easier for me to manage my Vitamin D intake. I make it a point to spend my breaks outdoors to benefit from natural sunlight.	0.1	4	Positive	0.4
17	I was amazed to learn about the role of Vitamin D in supporting the immune system, especially during pregnancy. This knowledge has motivated me to prioritize my Vitamin D intake through both diet and supplements.	0.125	7.8125	Positive	0.625
18	Finding vegetarian sources of Vitamin D has been a journey during my pregnancy. Fortified plant-based milk and UV-exposed mushrooms have become my go-to options.	0	0	Neutral	0
19	The first time I heard about the importance of Vitamin D in preventing preterm birth, I was taken aback. It has since become a focal point in my prenatal care routine, with regular monitoring and adjustments as needed.	0.125	2.564103	Positive	0.205128
20	Ever since I found out I’m deficient in Vitamin D during my first trimester, I’ve made it a point to walk in the park every morning. It’s not just about the vitamin; it’s my time to connect with nature and reflect on the journey ahead.	−0.075	3.875	Negative	0.516667
21	My nutritionist introduced me to a variety of Vitamin D-rich foods that I had never considered before. Incorporating mushrooms and fortified orange juice into my diet has been an interesting adventure.	0.5	25	Positive	0.5

## Data Availability

Not applicable.

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
