# Peer review of "Innovative Machine Learning Strategies for Early Detection and Prevention of Pregnancy Loss: The Vitamin D Connection and Gestational Health"

_diagnostics, 2024, doi:10.3390/diagnostics14090920_

Round 1
Reviewer 1 Report
Comments and Suggestions for Authors
The main focus of this paper is to explore the potential of using machine learning algorithms to predict and prevent early pregnancy loss, focusing on maternal serum vitamin D levels and other demographic, obstetric and anthropometric variables. It was found that machine learning models can accurately predict early pregnancy loss and include vitamin D levels as one of the important predictors. This manuscript has certain research values. However, for further improvement, here are some suggestions:
1. This manuscript has some obvious formatting problems:
l On page 4, lines 190 and 191 of the paper, there is a formatting problem, I suggest the authors to adjust them.
l In the article, on page 10, line 237, there is a formatting problem. I suggest the authors to adjust it.
l In the article, on page 10, line 245, there is a redundant spacing, I suggest the authors to remove it.
l On page 10, line 244 of the paper, I suggest the authors to move to the top of the image.
2. Section 5.1 includes the ‘Model Performance’ on Training Dataset, Test Dataset, and their ‘Performance analysis’ , so I title of Section 5.1 is not appropriate.
3. Please redraw Figure for a clear presentation.
4. Owing to loss of reference serial number, I was hard to follow this manuscript. I recommend that authors cite references in serial number form, just as the diagnostics template does.
5. The dataset was limited (68 participants were selected), which may affect the generalization of the model. In order to obtain more reliable results, I suggest the authors to validate the findings by extending datasets or adding new testing dataset.
6. I recommended that the authors added more details for model training (such as learning rate). Epoch alone is not enough.
7. The overfitting seems obvious. The authors should explain this phenomenon, or make some improvements (such as dropout).
8. This work uses machine learning as its main tool, so I think some novel work is worth citing:
l A guide to machine learning for biologists, Nature reviews moleclular cell biology, 2021.
l Understanding the brain with attention: a survey of transformers in brain sciences, Brain-x, 2023.
l A Neural Regression Model for Predicting Thermal Conductivity of CNT Nanofluids with Multiple Base Fluids, Journal of thermal science, 2021.
l Machine learning-enabled retrobiosynthesis of molecules, Nature catalysis, 2023.
Comments on the Quality of English LanguageThe English language requires appropriate modifications.
Author Response
Dear Reviewer,
Please see attached pdf where I addressed all your comments

Reviewer 2 Report
Comments and Suggestions for Authors
The aim of the paper titled as "Innovative Machine Learning Strategies for Early Detection and Prevention of Pregnancy Loss: The Vitamin D Connection and Gestational Health" is to explore the effectiveness of advanced machine learning algorithms in detecting and preventing early pregnancy loss , focusing on analyzing the role of maternal serum vitamin D levels and other relevant factors.
Specifically, it aims to develop and validate machine learning models to predict early pregnancy loss models using demographic, obstetric, anthropometric, and biochemical variables, with an emphasis on serum vitamin D levels, and validate these models using real-world clinical data to assess their accuracy and reliability.
depending on the experimental results, authors comapre the accuracy values.
The topic is interesting and it fits in the content of the Diagnostics Journal.
I have four strong and some small objections about the acceptance of the paper.
Mainly
"Table 24. Classical Models Name and Accuracy" values as
Model Accuracy
Logistic Regression 0.50
Random Forest 0.71
Support Vector Machine 0.64
Naive Bayes 0.57
K-Nearest Neighbors 0.50
are not acceptable.
I strongly suggest to check the implemented algotihms again.
0.50 accuracy means o learning in fact.
Therefore please check the model step buy step.
Secondly.
There is no data about the dataset.
I want to see its distribution and about its features.
Thirdly
What is the testing results?
I want to see the confusion matrix
Fourthly
If the size of dataset is small it is not reasonable to use deep learning.
Why do you prefer this approach?
Small ones are:
Line 190-191 are nor clear in the paper. they should be correted.
Figure 1 is not clear. it should be redrawn.
"4.1. Potential Metrics List" and "Table 13. Potential Metrics List" -- what is the effect of this table? It should be clear.
Classical model result evaluated by confusion matrix technique--Classical model results were evaluated by confusion matrix technique
and Advanced machine models were--??
Key determinants of EPL were identified, including levels of maternal serum vitamin D. -- ?
There are some potential metircs in Table 13 those are at dataset for further analysis.--?
Comments on the Quality of English LanguageIt can be acceptable.
There are some asmall errors. I have listed them.
Author Response
Dear Reviewer,
Enclosed with this correspondence, please find the revised manuscript in PDF format, and reply for reviewer comments which includes detailed responses to each of the comments you have provided. For your convenience, changes made in the manuscript are highlighted within the document.
Thank you for your consideration.
Best regards,
First Author
Md Abu Sufian

Reviewer 3 Report
Comments and Suggestions for Authors
This paper addresses a very important problem. Experiments are well defined and presented in detail. Domain is well described. Preliminary statistical analysis is sound. The problem with this paper is with ML. I suggest to invite a co-author knowledgable in applying ML to health problems.
ML/results section is not well organized, evaluation results section is unclear, some content is meaningless. What is shown in Table 22 is not a confusion matrix. One does not need to apply a range of classical ML methods, once it is established that a correlation exists, a single generally accepted methoud would do. Now clinical foundation models is the most popular for textual data; I suspect it would work fine in this domain as well
Author Response
Dear Reviewer,
Enclosed with this correspondence, please find the revised manuscript in PDF format and reply for reviewer comments which includes detailed responses to each of the comments you have provided. For your convenience, changes made in the manuscript are highlighted within the document.
Thank you for your consideration.
Best regards,
First Author
Md Abu Sufian

Round 2
Reviewer 2 Report
Comments and Suggestions for Authors
The authors have made the necessary corrections in response to previous reviews. It can be accepted as is.
Comments on the Quality of English LanguageIt can be accepted.